# Multi-Functional Isolated Three-Port Bidirectional DC/DC Converter for Photovoltaic Systems

**Yu-En Wu** * and **Rui-Ru Hong**

Department of Electronic Engineering, National Kaohsiung University of Science and Technology, Kaohsiung 811213, Taiwan
* Correspondence: yew@nkust.edu.tw; Tel.: +886-7-6011000 (ext. 32511)

**Abstract:** This paper proposes a novel multi-function isolated three-port bidirectional DC–DC converter for a stand-alone photovoltaic (PV) system. The proposed topology was composed of a unidirectional step-up converter and a bidirectional step-up/step-down converter that only required one set of complementary PWM signals to control any operation mode and used multiple operating stages to improve the practicability of the converter. In addition, the proposed topology had the function of inductance energy leakage recovery to improve the conversion efficiency and used synchronous rectification technology to reduce the conduction losses from passive components. This paper implemented a 500 W converter to verify the feasibility of the proposed converter by theoretical analysis, simulation, and experiment results. The experimental results show the highest efficiency of 95.5% for the PV step-up to the DC bus, 97.8% for the PV step-down to the battery terminal, 94.5% for the battery terminal step-up to the DC bus, and 93.4% for the DC bus step-down to the battery terminal, respectively.

**Keywords:** three-port bidirectional converter; leakage energy recycling; galvanic isolation; photovoltaic system

## 1. Introduction

Today's society relies heavily on non-renewable energy, which has seriously affected the environment [1]. The supply of petrochemical energy is excessively concentrated in areas with many disputes in the world, and energy prices are easily affected by human factors. Once oil production is in short supply, it will inevitably lead to global impacts such as soaring oil prices, rising prices, and inflation. In the case of natural gas, burning natural gas produces methane, a greenhouse gas that contributes to rapid global warming when it spills into the atmosphere in large quantities. Many countries are actively developing green energy sources, such as solar energy and wind power. Solar power generation has been the focus of energy technology in response to air pollution problems, which have received increased attention. On the other hand, once a solar power generation device is set up, energy can be continuously delivered, and there is no need to consider the problem of fuel transportation. Similarly, wind power generation does not need to consider the harmful gases and waste residues produced by fossil fuels such as coal and petroleum [2]. With the installation and promotion of wind power generation and solar power generation, centralized energy generation has gradually turned into distributed energy generation, which has become an inevitable trend of future urban intelligence [3]. With the reduction in the cost of solar cells, the number of commercialized solar power supply systems is increasing day by day. At the present stage, most common PV power generation systems are equipped with energy storage systems and rely on bidirectional converters to process the direction of energy and allow the energy storage system to be charged and discharged. This has led to the trend of research on bidirectional converters in recent years. Bidirectional converters can achieve step-up and step-down adjustments through the same topology.

Some scholars have successfully developed three-port converters for PV power supply systems, which can be powered by PV energy and batteries. Combining the functions of two-way converters and three-port converters is bound to greatly reduce the production cost and volume. Therefore, a three-port bidirectional converter with multiple operational functions was proposed in this paper. The PV could be stepped up to a DC bus (400 V) for supplying the DC load and charging the battery at the same time, while the battery also could be stepped up to the DC bus to stabilize the entire system. In addition, the DC bus could also be stepped down to charge the battery, as shown in Figure 1.

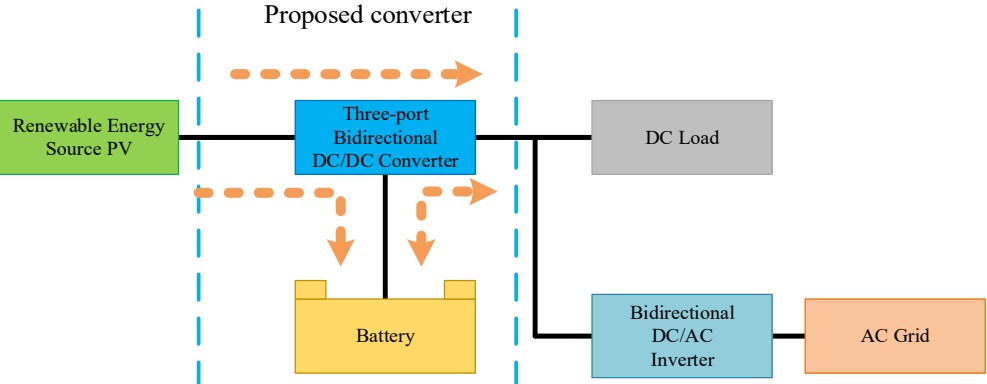

**Figure 1.** Application system diagram of three-port bidirectional DC/DC converter.

Bidirectional DC–DC converters can be divided into non-isolated and isolated types. Non-isolated converters [4–8] are derived from conventional converters. This type of converter has the advantages of a lower cost and a simpler structure. However, the voltage gain of the topology is not sufficient for high-voltage conversion applications. The converter [9] uses the switched capacitor voltage doubler technique, and the quasi-Z source can achieve a wider voltage gain range and lower power switch voltage stress. However, in order to achieve wide voltage gain, the switched capacitor technology increases the component count, resulting in lower conversion efficiency of this topology than conventional quasi-Z-source converters.

Refs. [10,11] presented the most common topology of all isolated converters, which have the advantages of few components, a simple structure, a low cost, and electrical isolation. However, leakage inductance can cause voltage surges to damage components, and therefore they should not be used in high-power applications. Ref. [11] presented a forward-flyback converter and a switched capacitor voltage circuit. The topology has the advantages of leakage inductance recovery, ZVS, and high conversion efficiency, and it can effectively improve the problem of large current ripples in traditional forward-flyback converters. However, circuit control is more complicated due to the synchronous rectification technology. At the same time, in order to achieve a high conversion ratio, a higher turns ratio is required, thus increasing the volume of the transformer and reducing the power density.

In recent years, many three-port converters have been proposed [12–21]. Such converters have been widely applied to renewable energy and electric vehicles, as they have fewer components and a lower cost. The topology in [12] requires larger component counts, two sets of transformers, and an inductor on the output side. Although there are two operating modes, the volume is large, which results in high implementation costs, leakage inductance, and low efficiency. Although [13] has the function of power factor correction (PFC) and has fewer active switching parts, it has the disadvantages of a discontinuous input current and low voltage gain. Ref. [14] was derived from the improved LLC topology, which uses a coupled inductor and a secondary-side voltage doubler circuit to achieve higher voltage gain and zero-voltage switching (ZVS). It also has the advantages of leakage inductance recovery and high efficiency. However, its operating modes are fewer. Ref. [15] presented a non-isolated converter with soft switching for hybrid energy systems. The

topology has the advantages of multiple input sources, high efficiency, a simple topology, and soft switching, but it possesses larger component counts and has fewer operating modes. Ref. [16] presented a non-isolated high-step-up three-port converter, in which coupled inductors are used to achieve high voltage gain. This topology possesses lower voltage stress, soft switching, and lower conduction losses, as it uses two sets of clamp circuits. Due to employing more active switches, the operating mode is more complicated. Ref. [17] employed two step-up circuits to achieve three-port bidirectional power flows, in which a clamp circuit is used to realize the soft switching. The design also reduces the number of components used and achieves higher voltage gain by combining two sets of voltage doubler rectifier circuits, thus satisfying the characteristics of high voltage gain, component sharing, and soft switching. However, four sets of magnetic components and eight capacitors are employed, resulting in a high cost and large volume. Ref. [18] designed a three-winding common-core coupled inductor combined with a full-wave frequency multiplier circuit to achieve a high step-up. This topology also has the advantages of primary and secondary leakage inductance recovery, high efficiency, a simple topology, and simple signal control. Only three switches are employed in [19], and two input sources share a single inductor, thus reducing size and cost. This topology also has the advantages of high voltage gain, low voltage stress, few active components, and simple control, and it does not require an excessive turns ratio or non-extreme duty cycle. However, there are fewer operating modes, and the topology is more complicated.

## 2. Circuit Architecture and Operational Principles

This paper proposed a three-port isolated bidirectional DC/DC converter for solar energy systems. Its circuit topology is shown in Figure 2a. The proposed topology mainly combines a unidirectional boost converter and a bidirectional step-up/step-down converter. The PV ($V_{PV}$) can be stepped up to a DC bus ($V_H$ 400 V) for supplying the DC load and charging the battery at the same time, and the battery ($V_{Bat}$) also can be stepped up to the DC bus. In addition, the DC bus can also be stepped down to charge the battery. This topology includes switches $S_1$–$S_6$, a transformer, an inductor ($L$), and capacitors $C_1$–$C_3$. The structure of this circuit is simple, and only one set of complementary signals is needed for control in different operating modes. It has leakage inductance recovery to reduce energy consumption, and it has four operating stages to achieve multiple operating functions under various conditions.

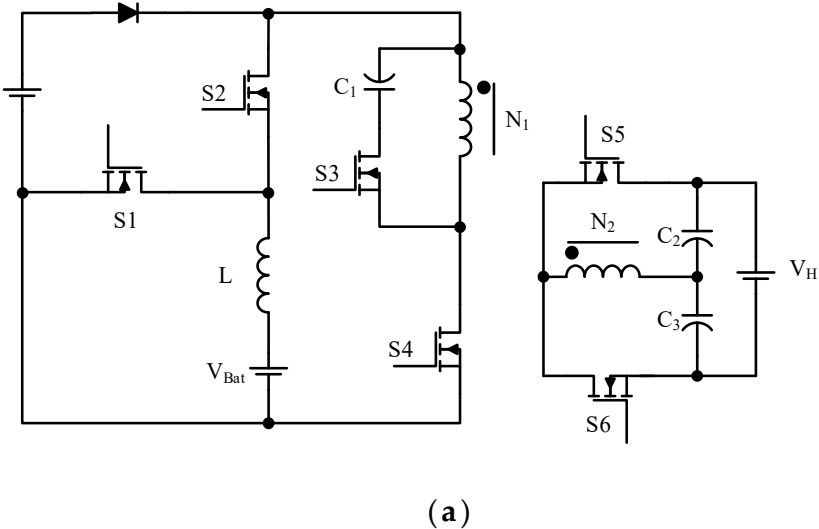

(**a**)

**Figure 2.** *Cont.*

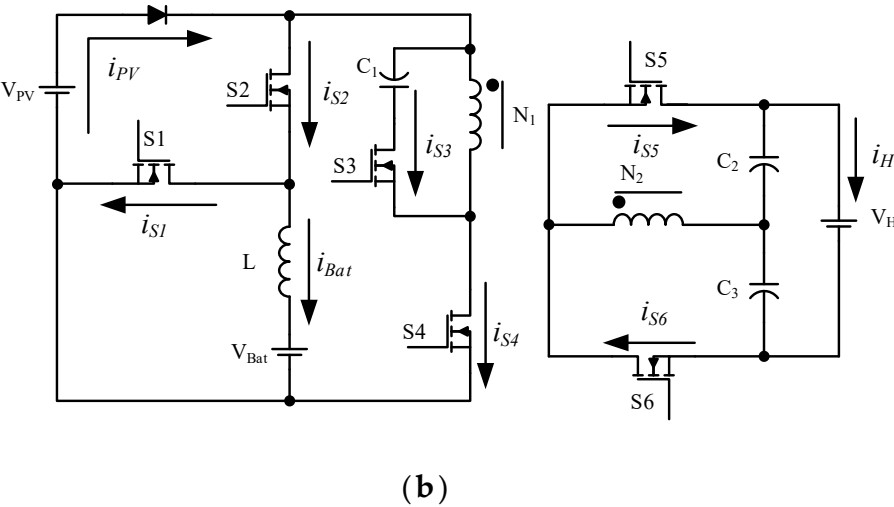

**(b)**

**Figure 2.** Proposed three-port bidirectional converter: (**a**) circuit architecture; (**b**) operating current direction of the proposed converter.

The operating current direction of the converter in this paper is shown in Figure 2b. The circuit was divided into four working modes for discussion and analysis. Stage one stepped up the output of PV ($V_{PV}$) to the high-voltage side ($V_H$) of the DC bus; stage two stepped down the output of the PV ($V_{PV}$) to charge the battery; stage three stepped up the battery output ($V_B$) to the high-voltage side ($V_H$); and stage four stepped down the high-voltage side ($V_H$) to charge the battery.

In order to simplify the analysis of the circuit, this study made the following assumptions:

(1)    Capacitors $C_{PV}$, $C_{Bat}$, and $C_1$–$C_3$ are large enough to keep the output voltage as a constant voltage source.
(2)    Switches $S_1$–$S_6$ are ideal components.
(3)    The inductor current is operated in a continuous conduction mode (CCM).
(4)    The switching period is T; the time when the switch turns on is $D$T, and the time when the switch turns off is $(1-D)$T.

### 2.1. Stage One: PV Output Is Stepped Up to the High-Voltage Side ($V_H$) of the DC Bus

Figure 3 shows the waveforms of the key components operating in stage one. In this stage, switches *S1* and *S2* are both turned off, and the energy is transmitted through the body diodes of switches *S5* and *S6*. This operation is divided into five modes in this stage, and the equivalent circuits are shown in Figure 4a–e.

(1)    Mode 1 [t0–t1]

At t = t0, the equivalent circuit is shown in Figure 4a. In this mode, switches *S3* and *S4* are turned off, and leakage inductance $L_k$ and capacitor $C_1$ store energy through the parasitic capacitance of switch *S3*. The parasitic capacitance of switch *S4* and magnetizing inductance $L_m$ transmit energy to the secondary side through the transformer. The energy of capacitor $C_2$ is transferred to the $V_H$ and charges capacitor $C_3$ at the same time. This mode ends when switch *S4* is turned on.

(2)    Mode 2 [t1–t2]

When t = t1, the equivalent circuit is shown in Figure 4b. In this mode, switch *S4* is turned on, and switch *S3* is turned off. The PV output ($V_{PV}$) stores energy in leakage inductance $L_k$ and magnetizing inductance $L_m$ and transfers the energy to the secondary side through the transformer for charging capacitor $C_2$. Capacitor $C_3$ releases energy to the $V_H$ through the body diode of switch *S5* and the secondary side of the transformer. Mode 2 ends when switch *S4* is turned off.

(3)  Mode 3 [t2–t3]

At t = t2, the equivalent circuit is shown in Figure 4c. This mode is a dead time for switches *S3* and *S4*. At this time, the leakage inductance releases energy to the magnetizing inductance $L_m$, the transformer, the parasitic capacitance of switch *S4*, and capacitor $C_1$, and stores energy in capacitor $C_2$ through the body diode of switch *S5*. Capacitor $C_3$ releases energy to the $V_H$ through the body diode of switch *S5* and the secondary side of the transformer. Mode 3 ends when switch *S3* is turned on.

(4)  Mode 4 [t3–t4]

At t = t3, the equivalent circuit is shown in Figure 4d. In this mode, switch *S3* is turned on, and switch *S4* is turned off. Leakage inductance $L_k$ releases energy to the transformer and capacitor $C_1$, and the secondary side of the transformer stores energy in capacitor $C_3$ through the body diode of switch *S6*. Capacitor $C_2$ releases energy to the $V_H$ through the body diode of switch *S6* and the secondary side of the transformer. When the energy of leakage inductance $L_k$ drops to zero, mode 4 ends.

(5)  Mode 5 [t4–t5]

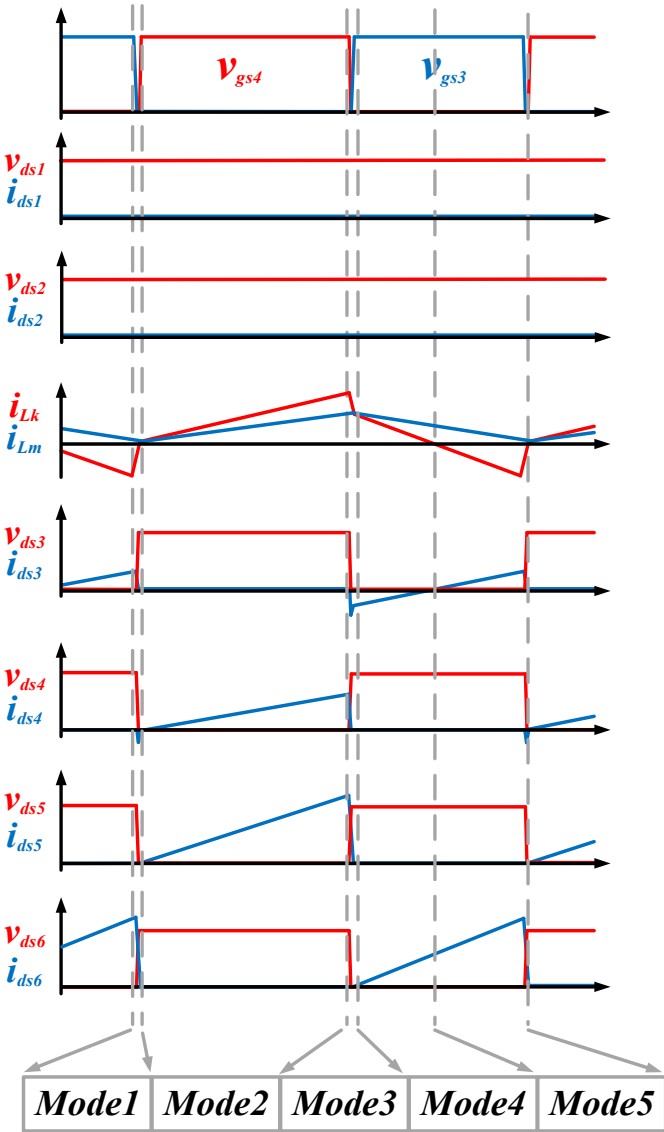

**Figure 3.** Waveforms of the key components in stage one.

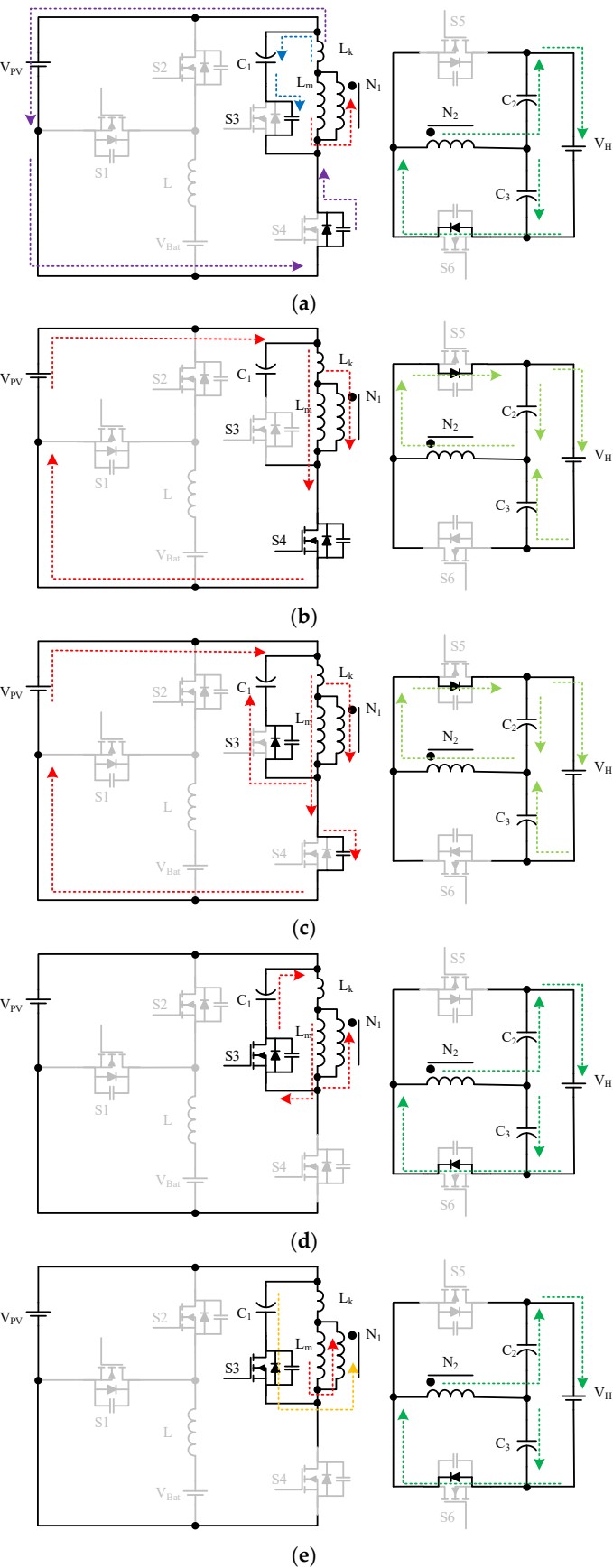

**Figure 4.** Operating equivalent circuits of stage one: (**a**) mode 1; (**b**) mode 2; (**c**) mode 3; (**d**) mode 4; and (**e**) mode 5.

At t = t4, the equivalent circuit is shown in Figure 4e. In this mode, switch S3 is turned on, and switch S4 is turned off. At this time, magnetizing inductance $L_m$ and capacitor $C_1$ release energy to the transformer, and the secondary side of the transformer stores energy in capacitor $C_3$ through the body diode of switch $S6$. Capacitor $C_2$ releases energy to the $V_H$ through the body diode of the switch and the secondary side of the transformer. Mode 5 ends when switch $S3$ is turned off.

### 2.2. Stage Two: PV Output Is Stepped Down to Charge the Battery

Figure 5 shows the waveforms of the key components in stage two. This stage is divided into two operating modes in one switching cycle. Switches $S3$–$S6$ are turned off. The equivalent circuits are shown in Figure 6a,b.

(1)  Mode 1 [t0–t1]

At t = t0, the equivalent circuit is shown in Figure 6a. In this mode, switch $S1$ is turned on, and switch $S2$ is turned off. Inductor $L$ releases energy to the battery ($V_{Bat}$) through switch $S1$. Mode 1 ends when switch $S1$ is turned off.

(2)  Mode 2 [t1–t2]

At t = t1, the equivalent circuit is shown in Figure 6b. In this mode, switch $S1$ is turned off, and $S2$ is turned on. PV output $V_{PV}$ stores energy in inductor $L$, and the energy of inductor $L$ charges the battery at the same time. Mode 2 ends when switch $S2$ is turned off.

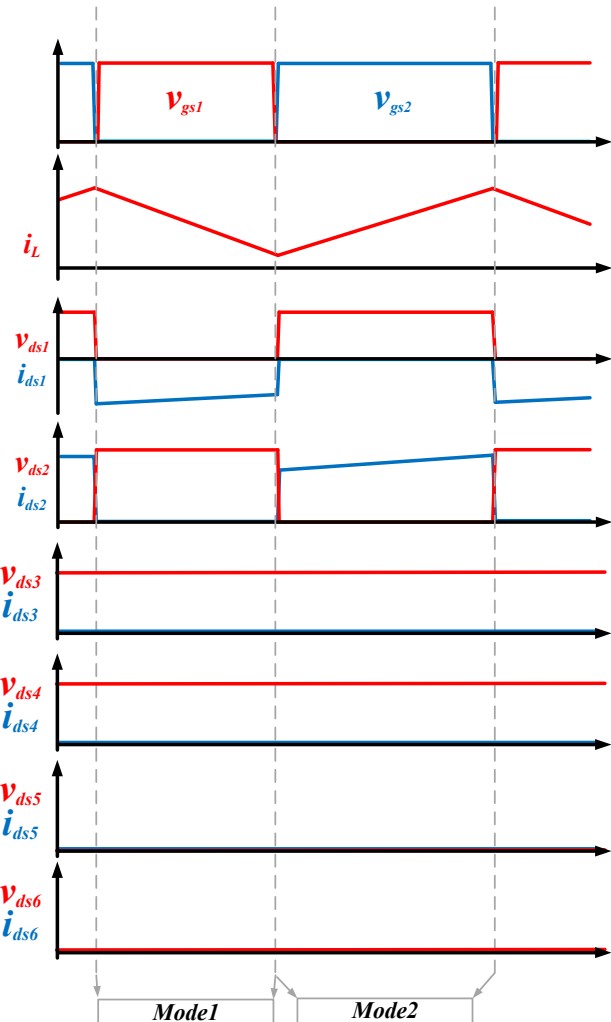

**Figure 5.** Waveforms of the key components in stage two.

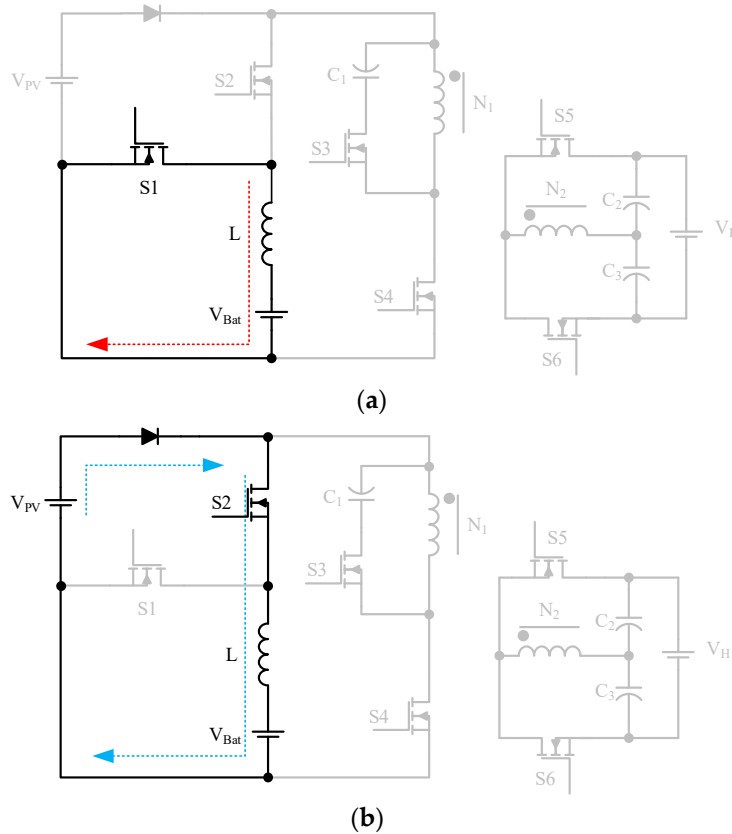

**Figure 6.** Operating equivalent circuits of stage two: (**a**) mode 1; (**b**) mode 2.

*2.3. Stage Three: Battery Output Is Stepped Up to the $V_H$ of the DC Bus*

Figure 7 shows the waveforms of the key components of stage three. This operating mode is divided into five modes in one switching cycle, and its equivalent circuits are shown in Figure 8a–e.

(1)    Mode 1 [t0–t1]

At t = t0, the equivalent circuit is shown in Figure 8a. In this mode, switches *S1–S6* continue to be turned off from the previous mode. Leakage inductance $L_k$ and the parasitic capacitance of switch *S3* release energy to capacitor $C_1$, capacitor $C_2$ stores energy through the secondary side of the transformer, and capacitor $C_3$ releases energy to the $V_H$ through the secondary side of the transformer. Mode 1 ends when switches *S1* and *S3* are turned on.

(2)    Mode 2 [t0–t1]

At t = t1, the equivalent circuit is shown in Figure 8b. In this mode, switches *S1* and *S3* are turned on, and switches *S2* and *S4* are turned off. At this time, the battery terminal ($V_{Bat}$) stores energy in inductor *L*, and leakage inductance $L_k$ releases energy to capacitor $C_1$ to recover the leakage inductance energy. Magnetizing inductance $L_m$ stores energy in capacitor $C_3$ through the secondary side of the transformer, and capacitor $C_2$ releases energy to the $V_H$ through the body diode of switch *S6* and the secondary side of the transformer. Mode 2 ends when the energy of leakage inductance $L_k$ is zero.

(3)    Mode 3 [t2–t3]

At t = t2, the equivalent circuit is shown in Figure 8c. In this mode, switches *S1–S4* are kept in the previous mode. At this time, leakage inductance $L_k$ and capacitor $C_1$ release energy to magnetizing inductance $L_m$ and transfer it to the secondary side of the transformer. The remaining operating states are the same as in the previous mode. Mode 3 ends when switches *S1* and *S3* are turned off.

(4)　Mode 4 [t3–t4]

At t = t3, the equivalent circuit is shown in Figure 8d. In this mode, all switches are turned off. Magnetizing inductance $L_m$ releases energy to capacitor $C_3$ through the secondary side of the transformer and the body diode of switch $S6$, and capacitor $C_2$ releases energy to the $V_H$ through the body diode of switch $S6$ and the secondary side of the transformer. Mode 4 ends when switches $S2$ and $S4$ are turned on.

(5)　Mode 5 [t4–t5]

At t = t4, the equivalent circuit is shown in Figure 8e. In this mode, switches $S2$ and $S4$ are turned on. Battery terminal $V_{Bat}$ and inductor $L$ release energy to the transformer and store energy in magnetizing inductor $L_m$. The secondary side of the transformer transfers energy to capacitor $C_2$ through the body diode of switch $S5$, and capacitor $C_3$ releases energy to the $V_H$ through the body diode of switch $S5$ and the secondary side of the transformer. Mode 5 ends when switches $S2$ and $S4$ are turned off.

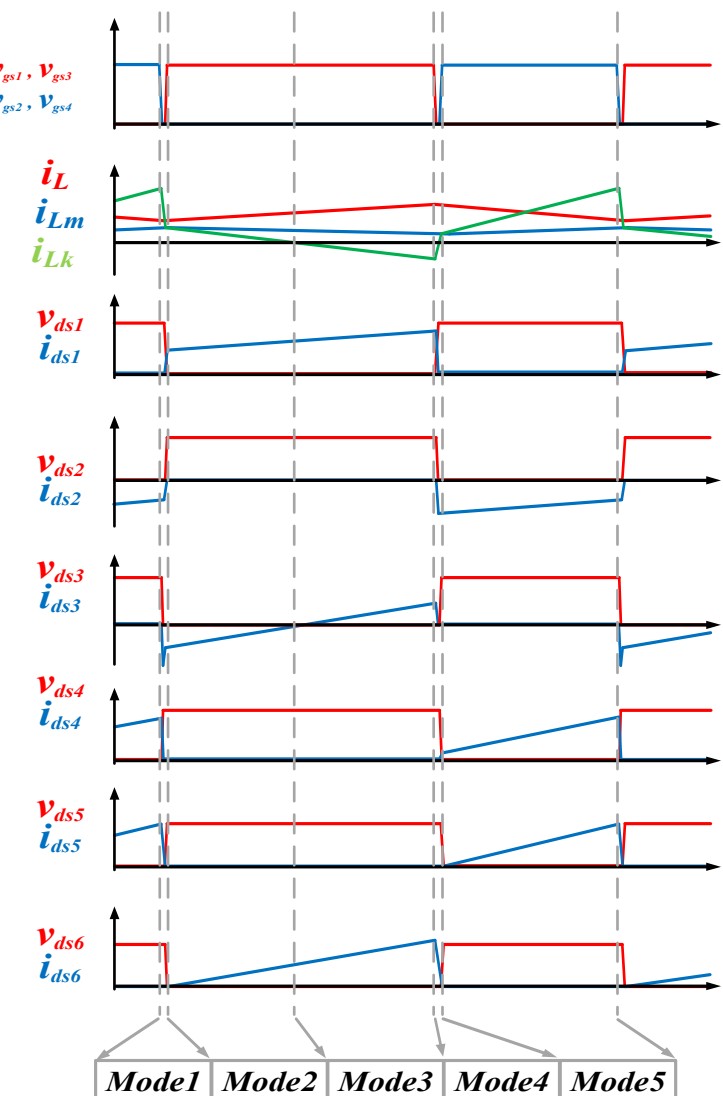

**Figure 7.** Waveforms of the key components in stage three.

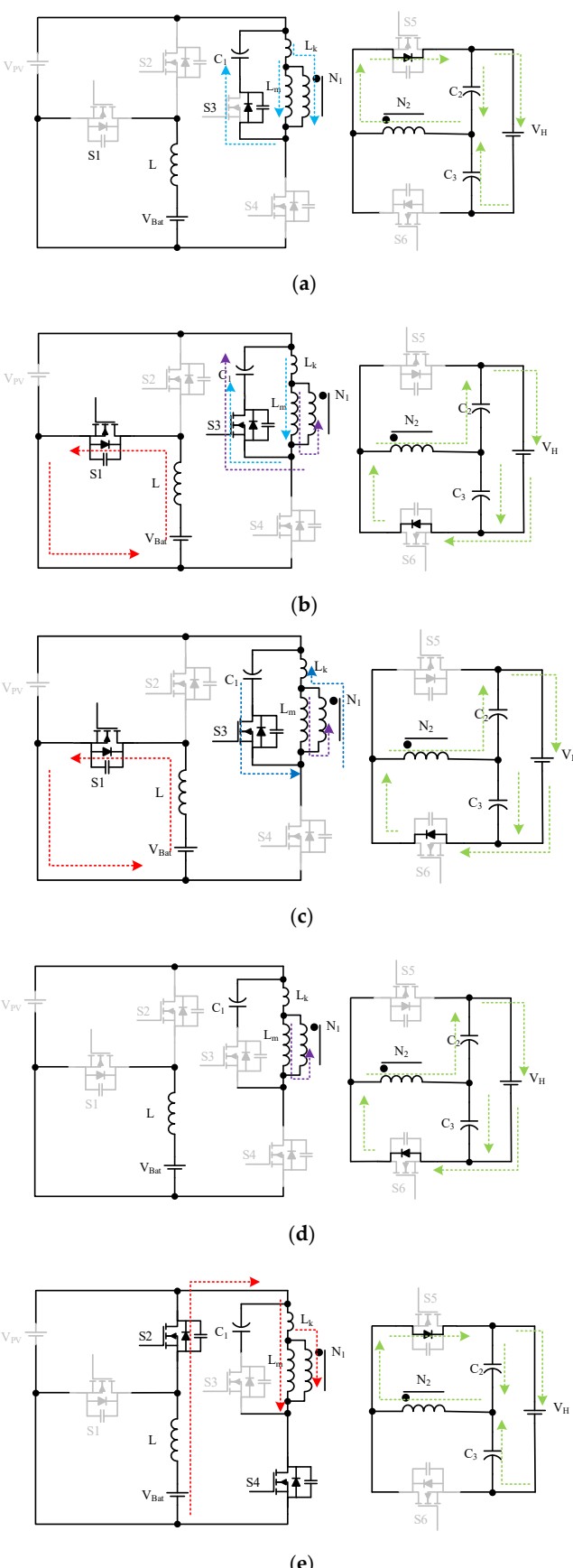

**Figure 8.** Operating equivalent circuits of stage three: (**a**) mode 1; (**b**) mode 2; (**c**) mode 3; (**d**) mode 4; and (**e**) mode 5.

### 2.4. Stage Four: The High-Voltage Side ($V_H$) Is Stepped Down to Charge the Battery

Figure 9 shows the waveforms of the key components in stage four. This operating mode is divided into six modes in one switching cycle, and the equivalent circuits are shown in Figure 10a–f.

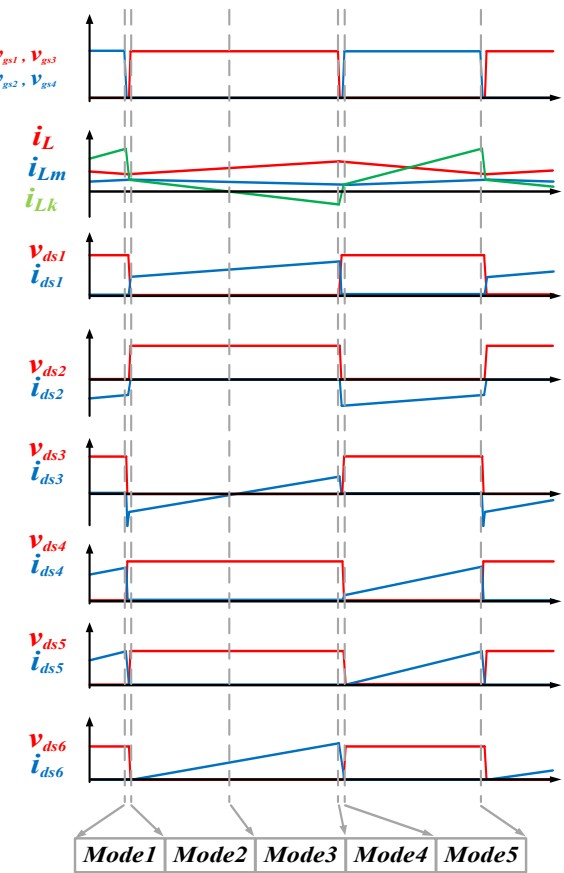

**Figure 9.** Waveforms of the key components in stage four.

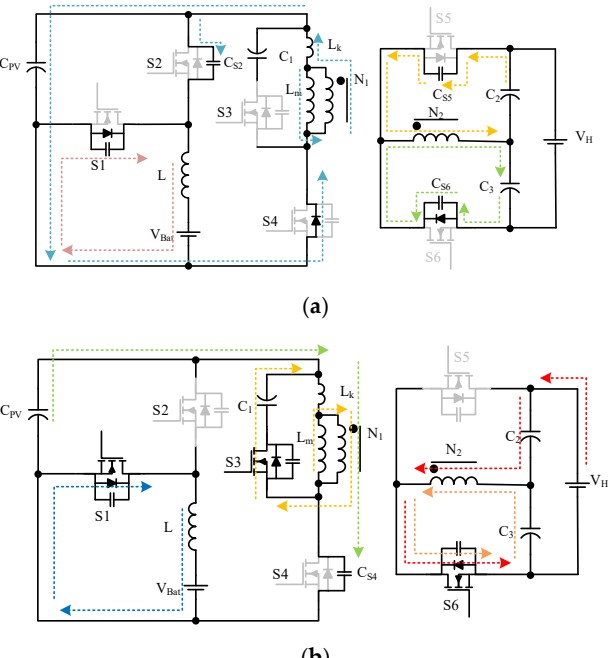

**Figure 10.** *Cont.*

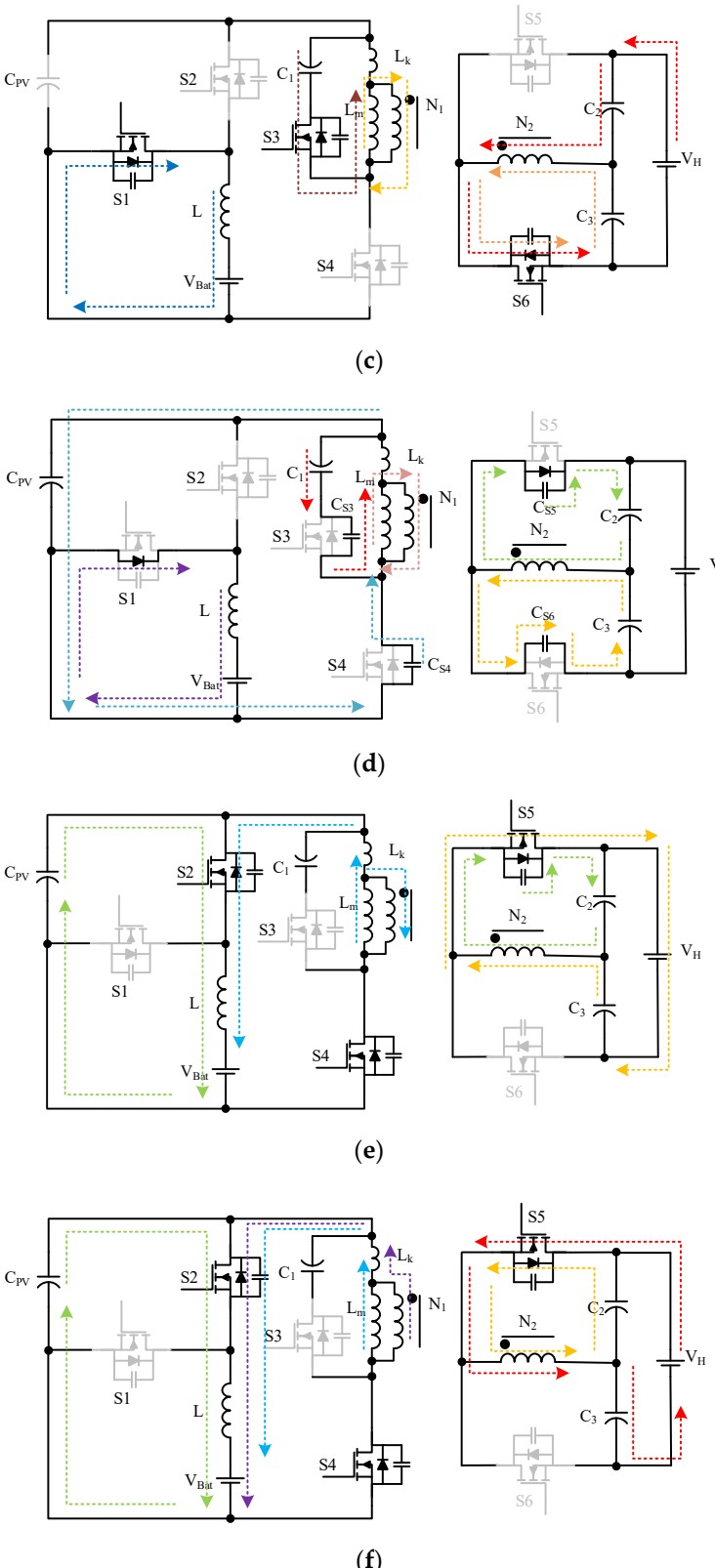

**Figure 10.** Equivalent circuit diagrams of stage four: (**a**) mode 1; (**b**) mode 2; (**c**) mode 3; (**d**) mode 4; (**e**) mode 5; and (**f**) mode 6.

(1)  Mode 1 [t0–t1]

At t = t0, the equivalent circuit is shown in Figure 10a. In this mode, switches *S1–S6* are all turned off following the previous mode. Capacitor $C_2$ releases energy to parasitic capacitor $C_{S5}$ of switch *S5*, while parasitic capacitor $C_{S6}$ of switch *S6* releases energy to capacitor $C_3$. Leakage inductance $L_k$ transfers energy to parasitic capacitance $C_{S2}$ of switch *S2* and the input capacitance ($C_{PV}$) of the PV terminal through the body diode of switch *S4*. Mode 1 ends when switches *S1*, *S3*, and *S6* are turned on.

(2)  Mode 2 [t0–t1]

At t = t1, the equivalent circuit is shown in Figure 10b. Switches *S1*, *S3*, and *S6* are turned on in this mode. The $V_H$ transfers energy to capacitor $C_2$ and the secondary side of the transformer, while capacitor $C_3$ also transfers energy to the primary side through the secondary side of the transformer. Leakage inductance $L_k$ releases energy to capacitor $C_1$ for leakage inductance recovery, magnetizing inductance $L_m$ stores energy through the primary side of the transformer while capacitor $C_{PV}$ transfers energy to parasitic capacitor $C_{S4}$ of switch *S4*, and inductor $L$ releases energy to the battery terminal $V_{Bat}$. Mode 2 ends when the energy of leakage inductance $L_k$ is zero.

(3)  Mode 3 [t2–t3]

At t = t2, the equivalent circuit is shown in Figure 10c. Switches *S1*, *S3*, and *S6* are turned on in this mode. At this time, leakage inductance $L_k$ and capacitor $C_1$ release energy to magnetizing inductance $L_m$. The other operating state is the same as in the previous Mode. Mode 3 ends when switches *S1*, *S3*, and *S6* are turned off.

(4)  Mode 4 [t3–t4]

At t = t3, the equivalent circuit is shown in Figure 10d. All switches are turned off in this mode. Capacitor *C3* discharges energy to parasitic capacitor $C_{S6}$ of switch *S6*, parasitic capacitor $C_{S5}$ of switch *S5* transfers energy to capacitor $C_2$, parasitic capacitor $C_{S4}$ of switch *S4* transfers energy to capacitor $C_{PV}$, capacitor $C_1$ releases energy to parasitic capacitance $C_{S3}$ of switch *S3*, and inductor $L$ releases energy to the battery terminal $V_{Bat}$ through the body diode of switch *S1*. Mode 4 ends when switches *S2*, *S4*, and *S5* are turned on.

(5)  Mode 5 [t4–t5]

At t = t4, the equivalent circuit is shown in Figure 10e. Switches *S2*, *S4*, and *S5* are turned on in this mode. Capacitor $C_2$ is charged by the transformer, capacitor $C_3$ transfers energy through the secondary side to the primary side of the transformer, and inductor $L$ stores energy through magnetizing inductor $L_m$ and capacitor $C_{PV}$. Mode 5 ends when the current in magnetizing inductance $L_m$ is greater than that of leakage inductance $L_k$.

(6)  Mode 6 [t5–t6]

At t = t5, the equivalent circuit is shown in Figure 10f. Switches *S2*, *S4*, and *S5* are turned on in this mode. Capacitor $C_2$ transfers energy to the primary side through the secondary side of the transformer, the $V_H$ stores energy in capacitor $C_3$ and at the same time transfers energy to the primary side through the secondary side of the transformer, and inductor $L$ stores energy through magnetizing inductor $L_m$, capacitor $C_{PV}$, and the primary side of the transformer. Mode 6 ends when switches *S2*, *S4*, and *S5* are turned off.

## 3. Steady-State Analysis

This section discusses and analyzes the voltage gain ratio and switching component stress, respectively. In order to simplify the analysis of the circuit, the same assumptions were also made, as in the above section.

### 3.1. Voltage Gain Analysis

(1)   Stage One

In stage one, the PV output $V_{PH}$ is stepped up to the $V_H$ when switch *S3* is turned on and switch *S4* is turned off, and the body diode of switch *S5* is reverse biased. According to Kirchhoff's voltage law, the voltage across magnetizing inductance Lm is as follows:

$$V_{L_m} = L_m \frac{di_{Lm}}{dt} = \frac{-V_{C3}}{n} \tag{1}$$

The inductor current decreases linearly with time when switch *S3* is closed. The variation in the inductor current with time is calculated as:

$$\frac{\Delta i_{L_m}}{\Delta t} = \frac{\Delta i_{L_m}}{DT} = \frac{-V_{C3}}{nL_m} \tag{2}$$

$(\Delta i_{Lm})_{\text{on}}$ can be derived as follows when switch *S3* is turned on:

$$(\Delta i_{L_m})_{on} = \frac{-V_{C3}DT}{nL_m} \tag{3}$$

When switch *S3* is turned off and *S4* is turned on, the PV terminal stores energy in magnetizing inductance $L_m$, and the voltage across the inductance is:

$$V_{L_m} = V_{PV} = L_m \frac{di_{Lm}}{dt} = \frac{V_{C2}}{n} \tag{4}$$

Similarly, the inductor current changes linearly with time when switch *S3* is turned off. The variation in inductor current with time is as follows:

$$\frac{\Delta i_{Lm}}{\Delta t} = \frac{\Delta i_{Lm}}{(1-D)T} = \frac{V_{PV}}{L_m} \tag{5}$$

$$(\Delta i_{L_m})_{off} = \frac{V_{PV}(1-D)T}{L_m} \tag{6}$$

According to the voltage-second balance, the variation in the inductor current must be zero in a steady-state operation:

$$(\Delta i_{L_m})_{on} + (\Delta i_{L_m})_{off} = 0 \tag{7}$$

$$\frac{-V_{C3}DT}{nL_m} + \frac{V_{PV}(1-D)T}{L_m} = 0 \tag{8}$$

After simplifying Equation (8), the voltage gain of $V_{PV}$ and $V_{C3}$ can be obtained as:

$$\frac{V_{C3}}{V_{PV}} = \frac{(1-D)}{D}n \tag{9}$$

Because:

$$V_H = V_{C2} + V_{C3} \tag{10}$$

After substituting (3) and (6) into (7), the voltage gain of the high-voltage side and the PV port can be obtained:

$$\frac{V_H}{V_{PV}} = \frac{n}{D} \tag{11}$$

According to Equation (11), the relationship between duty cycle *D*, turns ratio *n*, and the voltage gain ($V_H/V_{PV}$) of stage one can be obtained, as shown in Figure 11. Among them, the turns ratio *n* and duty cycle *D* used in this paper were 4 and 0.33, respectively; therefore, the voltage gain ($V_H/V_{PV}$) was 12.12.

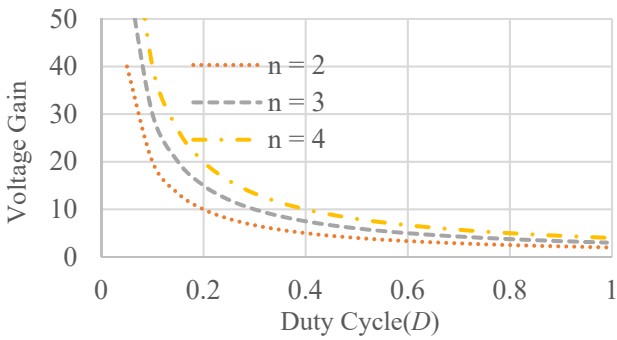

**Figure 11.** Relationship between the voltage gain ($V_H/V_{PV}$) of stage one and duty cycle D.

(2)    Stage Two

In stage two, the $V_{PH}$ is stepped down to charge the battery $V_{bat}$. When switch *S1* is turned on, the voltage across inductor *L* is:

$$V_L = V_{Bat} = L\frac{di_L}{dt} \tag{12}$$

$$\frac{\Delta i_L}{\Delta t} = \frac{\Delta i_{L_m}}{DT} = \frac{V_{Bat}}{L} \tag{13}$$

$(\Delta i_L)_{on}$ can be derived as follows when switch *S1* is turned on:

$$(\Delta i_L)_{on} = \left(\frac{V_{Bat}}{L}\right)DT \tag{14}$$

When switch *S1* is turned off and switch *S2* is turned on, the voltage across the inductor *L* is:

$$V_L = V_{Bat} - V_{PV} = L\frac{di_L}{dt} \tag{15}$$

$$\frac{di_L}{dt} = \frac{V_{Bat} - V_{PV}}{L} \tag{16}$$

At this time, $(\Delta i_L)_{off}$ is:

$$(\Delta i_L)_{off} = \left(\frac{V_{Bat} - V_{PV}}{L}\right)(1-D)T \tag{17}$$

According to the voltage-second balance, the variation in the inductor current must be zero in a steady-state operation:

$$(\Delta i_L)_{on} + (\Delta i_L)_{off} = 0 \tag{18}$$

$$\left(\frac{V_{Bat}}{L}\right)DT + \left(\frac{V_{Bat} - V_{PV}}{L}\right)(1-D)T = 0 \tag{19}$$

After simplifying Equation (19), the voltage gain in this stage can be obtained as:

$$\frac{V_{Bat}}{V_{PV}} = 1 - D \tag{20}$$

According to Equation (20), the relationship between duty cycle *D* and the voltage gain ($V_{Bat}/V_{PV}$) of stage two can be obtained, as shown in Figure 12. In this stage, duty cycle D was designed to be 0.5, and the voltage gain ($V_{Bat}/V_{PV}$) was 0.5.

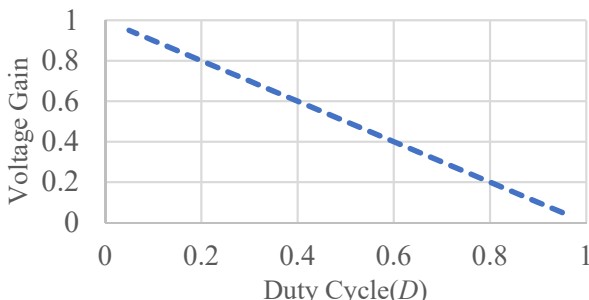

**Figure 12.** Relationship between the voltage gain ($V_{Bat}/V_{PV}$) of stage two and duty cycle D.

(3)　Stage Three

In this stage, the $V_{bat}$ is stepped up to the $V_H$. When switches *S1* and *S3* are turned on, the voltages of inductance *L* and magnetizing inductance *Lm* can be obtained by Kirchhoff's voltage law, respectively:

$$V_L = L\frac{di_L}{dt} = V_{Bat} \tag{21}$$

$$V_{Lm} = L_m\frac{di_{Lm}}{dt} = \frac{V_{C3}}{n} \tag{22}$$

When switches *S1* and *S3* are turned on and switches *S2* and *S4* are turned off, the inductor current increases linearly with time. The current variation of inductance *L* and magnetizing inductance $L_m$ with time can be calculated as follows:

$$\frac{\Delta i_L}{\Delta t} = \frac{\Delta i_L}{DT} = \frac{V_{Bat}}{L} \tag{23}$$

$$\frac{\Delta i_{Lm}}{\Delta t} = \frac{\Delta i_{Lm}}{DT} = \frac{V_{C3}}{L_m n} \tag{24}$$

$\Delta i_L$ and $\Delta i_{Lm}$ can be obtained as follows:

$$(\Delta i_L)_{on} = \frac{V_{Bat} DT}{L} \tag{25}$$

$$(\Delta i_{Lm})_{on} = \frac{V_{C3} DT}{L_m n} \tag{26}$$

When switches *S1* and *S3* are turned off and switches *S2* and *S4* are turned on, the inductor current does not change instantaneously. The voltage across inductance *L* and magnetizing inductance $L_m$ is:

$$V_L = (V_{Bat} - V_{Lm}) = L\frac{di_L}{dt} \tag{27}$$

$$V_{Lm} = \frac{-V_{C2}}{n} = L_m\frac{di_{Lm}}{dt} \tag{28}$$

When switches *S1* and *S3* are turned off, the current variation of inductance *L* and magnetizing inductance $L_m$ with time is:

$$\frac{\Delta i_L}{\Delta t} = \frac{\Delta i_L}{(1-D)T} = \frac{(V_{Bat} - V_{Lm})}{L} \tag{29}$$

$$(\Delta i_L)_{off} = \frac{(V_{Bat} - V_{Lm})(1-D)T}{L} \tag{30}$$

$$\frac{\Delta i_{Lm}}{\Delta t} = \frac{\Delta i_{Lm}}{(1-D)T} = \frac{-V_{C2}}{L_m n} \tag{31}$$

$$(\Delta i_{\text{Lm}})_{off} = \frac{-V_{C2}(1-D)T}{L_m n} \quad (32)$$

According to the voltage-second balance, the change in the inductor current must be zero in a steady-state operation:

$$(\Delta i_L)_{on} + (\Delta i_L)_{off} = 0 \quad (33)$$

$$\frac{V_{Bat}DT}{L} + \frac{(V_{Bat} - V_{Lm})(1-D)T}{L} = 0 \quad (34)$$

Simplifying (34), the relationship between $V_{Bat}$ and $V_L$ can be obtained as:

$$V_L = \frac{V_{Bat}}{1-D} \quad (35)$$

Similarly,

$$(\Delta i_{Lm})_{on} + (\Delta i_{Lm})_{off} = 0 \quad (36)$$

$$\frac{V_{C3}DT}{L_m n} + \frac{-V_{C2}(1-D)T}{L_m n} = 0 \quad (37)$$

Simplifying (37) obtains (38):

$$\frac{V_{C2} + V_{C3}}{n}D = \frac{V_{C2}}{n} \quad (38)$$

Because:

$$V_{C2} + V_{C3} = V_H \quad (39)$$

Substituting (28) and (38) into (39) obtains (40):

$$\frac{V_H}{n}D = L_m \quad (40)$$

Simplifying (34) and (40) obtains the voltage gain of stage three as follows:

$$\frac{V_H}{V_{Bat}} = \frac{n}{(1-D)D} \quad (41)$$

The relationship between duty cycle $D$, turns ratio $n$, and the voltage gain ($V_H/V_{Bat}$) of stage three can be obtained, as shown in Figure 13. Among them, the turns ratio n and duty cycle D designed in this paper were 4 and 0.5, respectively; therefore, the voltage gain ($V_H/V_{Bat}$) was 16.6.

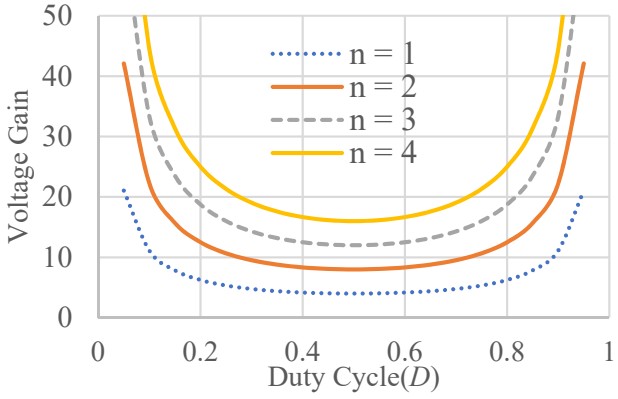

**Figure 13.** Relationship between the voltage gain ($V_H/V_{Bat}$) of stage three and duty cycle D.

(4)    Stage Four

In this stage, the $V_H$ is stepped down to charge the $V_{bat}$. When switches *S1*, *S3*, and *S6* are turned on, the voltages of inductance *L* and magnetizing inductance $L_m$ can be obtained from Kirchhoff's voltage law:

$$V_L = L\frac{di_L}{dt} = V_{Bat} \tag{42}$$

$$V_{Lm} = L_m\frac{di_{Lm}}{dt} = \frac{V_{C3}}{n} = \frac{V_H - V_{C2}}{n} \tag{43}$$

When switches *S1*, *S3*, and *S6* are turned on and switches *S2*, *S4*, and *S5* are turned off, the current increases linearly with time. The current variation of inductance *L* and magnetizing inductance $L_m$ with time can be calculated as:

$$\frac{\Delta i_L}{\Delta t} = \frac{\Delta i_L}{DT} = \frac{V_{Bat}}{L} \tag{44}$$

$$\frac{\Delta i_{Lm}}{\Delta t} = \frac{\Delta i_{Lm}}{DT} = \frac{V_{C3}}{L_m n} = \frac{V_H - V_{C2}}{L_m n} \tag{45}$$

$(\Delta i_L)_{on}$ and $(\Delta i_{Lm})_{on}$ can be obtained as follows:

$$(\Delta i_L)_{on=}\frac{V_{Bat}DT}{L} \tag{46}$$

$$(\Delta i_{Lm})_{on=}\frac{(V_H - V_{C2})DT}{L_m n} \tag{47}$$

When switches *S1*, *S3*, and *S6* are turned off and switches *S2*, S4, and *S5* are turned on, the inductor current does not change instantaneously. The voltage across inductance *L* and magnetizing inductance $L_m$ is:

$$V_L = (V_{Bat} - V_{Lm}) = \left(V_{Bat} - \frac{V_{C2}}{n}\right) = L\frac{di_L}{dt} \tag{48}$$

$$V_{Lm} = \frac{-V_{C2}}{n} = L_m\frac{di_{Lm}}{dt} \tag{49}$$

Similarly, when switches *S1*, *S3*, and *S6* are turned off, the current variation of inductance *L* and magnetizing inductance $L_m$ with time is:

$$\frac{\Delta i_L}{\Delta t} = \frac{\Delta i_L}{(1-D)T} = \frac{(V_{Bat} - V_{Lm})}{L} \tag{50}$$

$$(\Delta i_L)_{off} = \frac{(V_{Bat} - V_{Lm})(1-D)T}{L} \tag{51}$$

$$\frac{\Delta i_{Lm}}{\Delta t} = \frac{\Delta i_{Lm}}{(1-D)T} = \frac{-V_{C2}}{L_m n} \tag{52}$$

$$(\Delta i_{Lm})_{off} = \frac{-V_{C2}(1-D)T}{L_m n} \tag{53}$$

According to the voltage-second balance, the change in the inductor current must be zero in a steady-state operation:

$$(\Delta i_{Lm})_{on} + (\Delta i_{Lm})_{off} = 0 \tag{54}$$

$$\frac{(V_H - V_{C2})DT}{L_m} + \frac{-V_{C2}(1-D)T}{L_m} = 0 \tag{55}$$

Simplifying (55), the relationship between $V_{C2}$ and $V_H$ can be obtained as:

$$V_{C2} = V_H D \tag{56}$$

Similarly,

$$(\Delta i_L)_{on} + (\Delta i_L)_{off} = 0 \tag{57}$$

$$\frac{V_{Bat} DT}{L} + \frac{\left(V_{Bat} - \frac{V_{C2}}{n}\right)(1-D)T}{L} = 0 \tag{58}$$

Simplifying (58) obtains (59):

$$\frac{V_{Bat}}{V_{C2}} = \frac{1-D}{n} \tag{59}$$

By substituting (56) into (59) and simplifying the equation, the voltage gain of stage four can be obtained as:

$$\frac{V_{Bat}}{V_H} = \frac{(1-D)D}{n} \tag{60}$$

The relationship between duty cycle D, turns ratio $n$, and the voltage gain ($V_{Bat}/V_H$) of stage four can be obtained, as shown in Figure 14. Among them, the turns ratio $n$ and duty cycle D designed in this paper were 4 and 0.5, respectively; therefore, the voltage gain ($V_{Bat}/V_H$) was 0.0625. It could be seen that the voltage gain ($V_{Bat}/V_H$) could be increased by reducing duty cycle D or increasing turns ratio $n$.

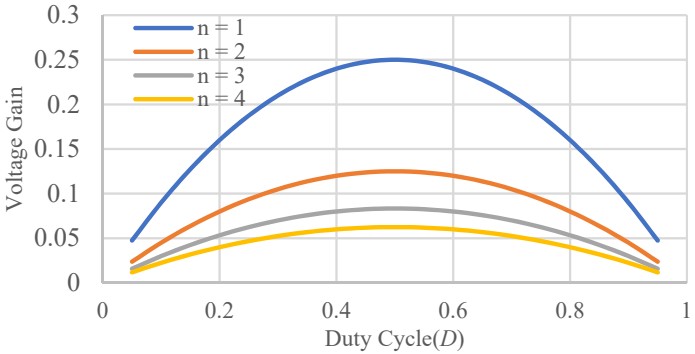

**Figure 14.** Relationship between the voltage gain ($V_{Bat}/V_H$) of stage four and duty cycle D.

### 3.2. Voltage Stress Analysis

The voltage stress on the components of the proposed topology can be deduced from the ON–OFF state of the switches. The voltage stress of switch *S1* can be derived from the ON state of mode 5 in stage four, as shown in Figure 10e and as follows:

$$V_{S1} - V_{Lm} = 0 \tag{61}$$

By substituting (49) and (56) into (61), the voltage stress of switch *S1* can be obtained:

$$V_{S1} = \frac{V_H D}{n} \tag{62}$$

The voltage stress of switch *S2* can be deduced from the state of mode 2 in stage four, as shown in Figure 10b and as follows:

$$V_{S2} = \frac{V_H D}{n} \tag{63}$$

The voltage stress of switch *S3* can be derived from the state of mode 5 in stage four, as shown in Figure 10e and as follows:

$$V_{S3} = \frac{V_H}{n} \tag{64}$$

The voltage stress of switch *S4* can be derived from the state of mode 3 in stage four, as shown in Figure 10c and as follows:

$$V_{S4} = \frac{V_H}{n} \tag{65}$$

The voltage stress of switch *S5* can be derived from the state of mode 2 in stage four, as shown in Figure 10b and as follows:

$$V_{S5} = V_H \tag{66}$$

The voltage stress of switch *S6* can be derived from the state of mode 5 in stage four, as shown in Figure 10e and as follows:

$$V_{S6} = V_H \tag{67}$$

The voltage stress of capacitor $C_{PV}$ is $V_{PV}$.

$$V_{CPV} = V_{PV} \tag{68}$$

The voltage stress of capacitor $C_{Bat}$ is $V_{Bat}$.

$$V_{CBat} = V_{Bat} \tag{69}$$

The voltage stress of capacitor $C_2$ is:

$$V_{C2} = V_H(1 - D) \tag{70}$$

The voltage stress of capacitor $C_3$ is:

$$V_{C3} = V_H D \tag{71}$$

*3.3. Magnetic Component Design*

Since the proposed converter operates in CCM, this section presents the design of inductance $L$ and magnetizing inductance $L_m$.

(1) Design of Inductor L

The maximum and minimum inductor current ($i_{Lmax}$ and $i_{Lmin}$) can be expressed as follows when switch *S1* is turned on and off:

$$i_{L(max)} = i_L + \frac{\Delta i_L}{2} \tag{72}$$

$$i_{L(min)} = i_L - \frac{\Delta i_L}{2} \tag{73}$$

Because:

$$i_L = i_{Bat} \tag{74}$$

According to the energy conservation principle, the ideal transformer conversion efficiency is 100%, and the input power ($P_{Bat}$) is equal to the output power ($P_H$):

$$P_{Bat} = P_H \tag{75}$$

$$V_{Bat} i_{Bat} = V_H i_H \tag{76}$$

By substituting (41) and (74) into (76) and replacing output current $i_H$ with output resistance $R_H$ on the high-voltage side, the equation can be simplified as follows:

$$i_L = \frac{V_{Bat}n^2}{[(1-D)D]^2 R_H} \tag{77}$$

Substituting (25) and (77) into (72) and (73) obtains:

$$i_{L(max)} = \frac{V_{Bat}n^2}{[(1-D)D]^2 R_H} + \frac{V_{Bat}DT}{2L} \tag{78}$$

$$i_{L(min)} = \frac{V_{Bat}n^2}{[(1-D)D]^2 R_H} - \frac{V_{Bat}DT}{2L} \tag{79}$$

Due to all of the inductances of the converters in this paper working in CCM, $i_L$ is always greater than zero. If the converter operates in the boundary conduction mode (BCM), $i_{L(min)}$ is equal to zero. Therefore, the boundary value between the continuous and discontinuous inductor current is expressed as:

$$i_{L(min)} = 0 = \frac{V_{Bat}n^2}{[(1-D)D]^2 R_H} - \frac{V_{Bat}DT}{2L} \tag{80}$$

According to the above equation, the value of inductance $L_{(BCM)}$ can be obtained by substituting switching frequency f and simplifying the equation:

$$L_{(BCM)} = \frac{(1-D)^2 D^3 R_H}{2fn^2} \tag{81}$$

It can be seen from (81) that the value of inductance $L$ can be designed by switching frequency $f$, transformer turns ratio $n$, and duty cycle ratio $D$ to determine whether the converter operates in BCM, CCM, or the discontinuous conduction mode (DCM). The turns ratio $n$ was set at 4 in this paper, and the switching frequency $f$ was 40 kHz. Substituting the above parameters into (81) obtains the inductance value, as shown in Figure 15. When the value of inductance L is greater than $L_{(BCM)}$, the converter will operate in CCM; conversely, when the value of inductance $L$ is lower than $L_{(BCM)}$, it will operate in DCM.

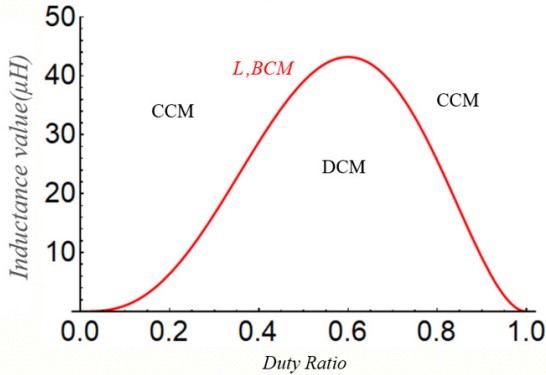

**Figure 15.** Inductor $L$ operating in the BCM.

(2)    Design of Magnetizing Inductance ($L_m$)

The maximum and minimum magnetizing inductor current ($i_{Lm(max)}$ and $i_{Lm(min)}$) can be expressed as follows when switch S4 is turned on and off:

$$i_{Lm(max)} = i_{Lm} + \frac{\Delta i_{Lm}}{2} \tag{82}$$

$$i_{Lm(min)} = i_{Lm} - \frac{\Delta i_{Lm}}{2} \tag{83}$$

Because:

$$i_{Bat} = i_{Lm}D \tag{84}$$

Similarly,

$$P_{Bat} = P_H \tag{85}$$

$$V_{Bat}i_{Bat} = V_H i_H \tag{86}$$

By substituting (41) and (84) into (86) and replacing output current $i_H$ with output resistance $R_H$ on the high-voltage side, the equation can be simplified as follows:

$$i_{Lm} = \frac{V_{Bat}n^2}{(1-D)^2 D^3 R_H} \tag{87}$$

Substituting (32) into (72) and (73) obtains:

$$i_{Lm(max)} = \frac{V_{Bat}n^2}{(1-D)^2 D^3 R_H} + \frac{V_{C2}(1-D)T}{2L_m n} \tag{88}$$

$$i_{Lm(min)} = \frac{V_{Bat}n^2}{(1-D)^2 D^3 R_H} - \frac{V_{C2}(1-D)T}{2L_m n} \tag{89}$$

If the converter operates in BCM, $i_{Lm(min)}$ is equal to zero. Therefore:

$$i_{Lm(min)} = 0 = \frac{V_{Bat}n^2}{[(1-D)D]^2 R_H} - \frac{V_{C2}(1-D)T}{2L_m n} \tag{90}$$

By substituting duty cycle T into (90) and simplifying the equation, magnetizing inductance $L_{m(BCM)}$ can be obtained:

$$L_{m(BCM)} = \frac{(1-D)^2 D^3 R_H}{2fn} \tag{91}$$

It can be seen that $L_{m(BCM)}$ can be determined by switching frequency $f$, transformer turns ratio $n$, and duty cycle ratio $D$. Figure 16 shows the relationship between magnetizing inductance $L_m$ and the duty ratio $D$. The proposed converter operates in CCM when the value of magnetizing inductance $L_m$ is greater than $L_{m(BCM)}$; conversely, when the value of magnetizing inductance Lm is lower than $L_{m(BCM)}$, it will operate in DCM.

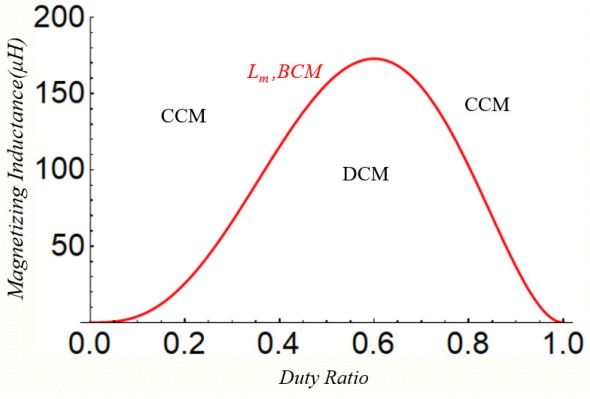

**Figure 16.** Magnetizing inductance Lm operating in the BCM.

## 4. Experimental Results

The electrical specifications and component parameters of the proposed converter are listed in Table 1. Figure 17 shows the implemented circuit in this paper. In this paper, the PV voltage was 48 V, the $V_{bat}$ was 24 V, and the $V_H$ was 400 V.

**Table 1.** Electrical specifications and component parameters of the proposed topology.

| Parameter | Specification |
|---|---|
| Input PV voltage $V_{PV}$ | 48 V |
| Battery voltage $V_{Bat}$ | 24 V |
| High-side voltgae $V_H$ | 400 V |
| Maximus output battery power | 500 W |
| Maximus output load bus power | 500 W |
| Switching frequency $fs$ | 40 kHz |

| Component | Model | Specification |
|---|---|---|
| $S_1$–$S_4$ | IRFP4568PbF | 150 V/171 A |
| $S_5$ and $S_6$ | IXFH60N50P3 | 500 V/60 A |
| $C_1$ | Electroytic Capacitor | 50 µF/100 V |
| $C_2$ and $C_3$ | Electroytic Capacitor | 330 µF/450 V |
| $L$ | MPP Ring Core | 40 µH |
| $Lm$ | MPP Ring Core | 170 µH |

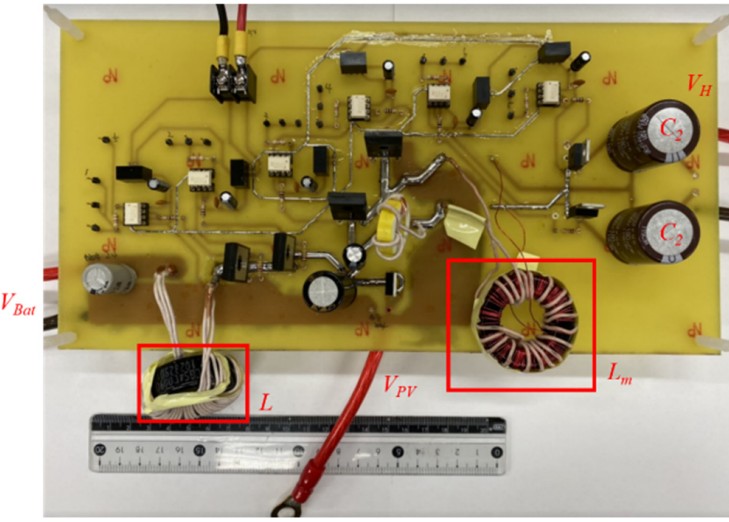

**Figure 17.** Photograph of the proposed converter.

Figure 18a–e shows the measured waveforms of stage one operating under a full load (500 W). Figure 18a shows the measured waveforms $v_{gs3}$, $v_{ds3}$, and $i_{ds3}$ of switch *S3*. It can be seen that when switch *S3* was turned on, $i_{ds3}$ started to rise from a negative to a positive current, leakage inductance current $i_{Lk}$ was released by switch *S3*, and the leakage inductance is recovered. Figure 18b shows the measured waveforms ($v_{gs4}$, $v_{ds4}$, and $i_{ds4}$) of switch *S4*. It can be seen that magnetizing inductance $L_m$ stored energy through switch *S4*. Figure 18c shows the measured waveforms $v_{gs4}$, $v_{ds5}$, and $i_{ds5}$. In this stage, in order to avoid circuit malfunctions caused by an unbalanced magnetic flux on the secondary side, the current flowed through the body diode of switch *S5*. Figure 18d shows the measured waveforms $v_{gs3}$, $v_{ds6}$, and $i_{ds6}$. The current flowed through the body diodes of switch *S6*. Figure 18e shows the measured waveform $v_{gs3}$, the primary-side current $i_{n1}$, and the secondary-side current $i_{n2}$ of the transformer.

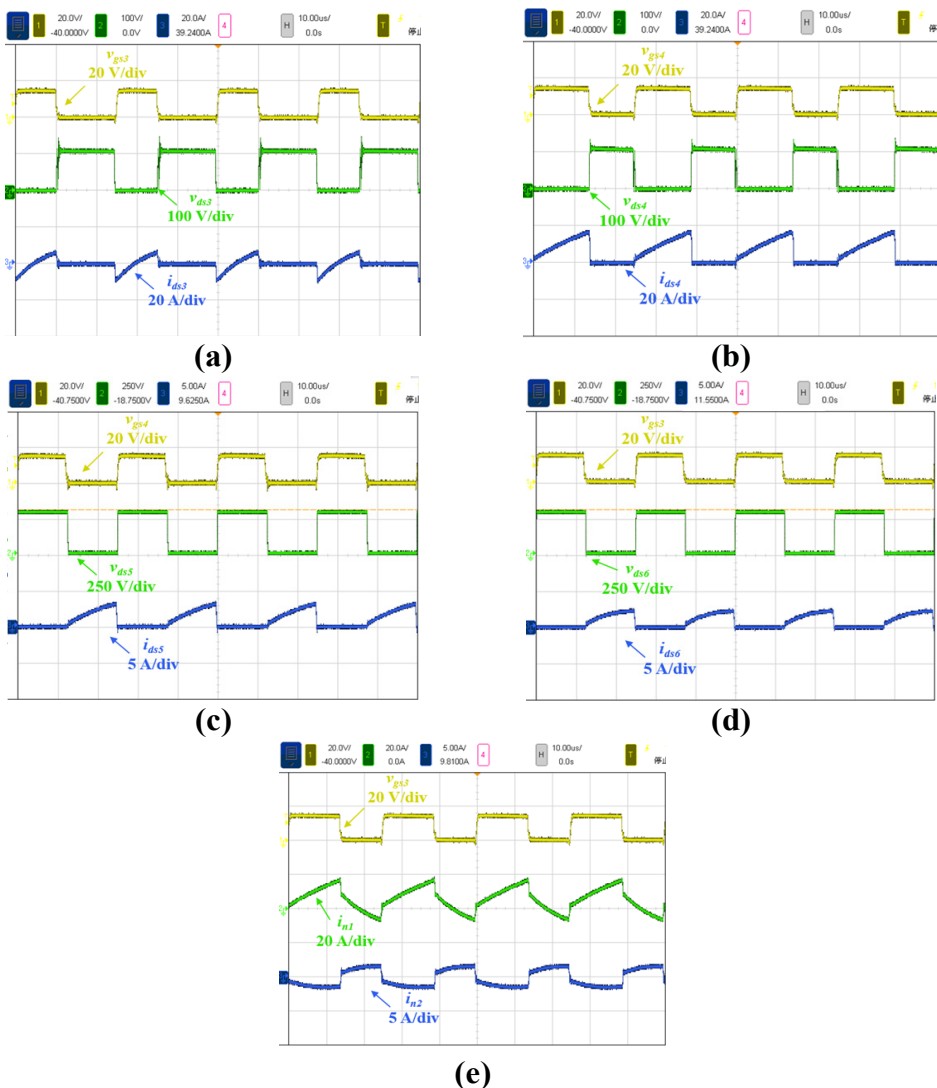

**Figure 18.** Measured waveforms of stage one: (**a**) $v_{gs3}$, $v_{ds3}$, and $i_{ds3}$; (**b**) $v_{gs4}$, $v_{ds4}$, and $i_{ds4}$; (**c**) $v_{gs4}$, $v_{ds5}$, and $i_{ds5}$; (**d**) $v_{gs3}$, $v_{ds6}$, and $i_{ds6}$; and (**e**) $v_{gs3}$, $i_{n1}$, and $i_{n2}$.

Figure 19a–c show the measured waveforms of stage two operating under a full load (500 W). Figure 19a shows the measured waveforms ($v_{gs1}$, $v_{ds1}$, and $i_{ds1}$) of switch *S1*. It can be seen that switch *S1* had synchronous rectification characteristics in this stage. Figure 19b shows the measured waveforms ($v_{gs2}$, $v_{ds2}$, and $i_{ds2}$) of switch *S2*. Figure 19c shows the measured waveforms of $v_{gs2}$, $v_{ds2}$, and $i_L$. It can be seen that inductor *L* stored energy while switch *S2* was turned on and released energy while switch *S2* was turned off.

Figure 20a–h show the measured waveforms of stage three operating under full load (500 W). Figure 20a shows the measured waveforms $v_{gs1}$, $v_{ds1}$, and $i_{ds1}$ of switch *S1*, and it can be seen that inductor *L* stored energy through switch *S1*. Figure 20b shows the measured waveforms ($v_{gs2}$, $v_{ds2}$, and $i_{ds2}$) of switch *S2*. Figure 20c shows the measured waveforms ($v_{gs3}$, $v_{ds3}$, and $i_{ds3}$) of switch *S3*. When switch *S3* was turned on, current $i_{ds3}$ started to turn from a negative current to a positive current, and leakage inductance current $i_{Lk}$ was released and recovered energy through switch *S3*. Figure 20d shows the measured waveforms ($v_{gs4}$, $v_{ds4}$, and $i_{ds4}$) of switch *S4*. Figure 20e shows the measured waveforms $v_{gs4}$, $v_{ds5}$, and $i_{ds5}$. Figure 20f is the measured waveforms $v_{gs3}$, $v_{ds6}$, and $i_{ds6}$. In this mode, the energy flowed through the body diode of switch *S6*. Figure 20g shows the measured waveform $v_{gs1}$, inductor current $i_L$, and the transformer primary-side current $i_{n1}$. Figure 20h shows the measured waveform $v_{gs1}$, the transformer primary-side current $i_{n1}$, and the transformer secondary-side current $i_{n2}$.

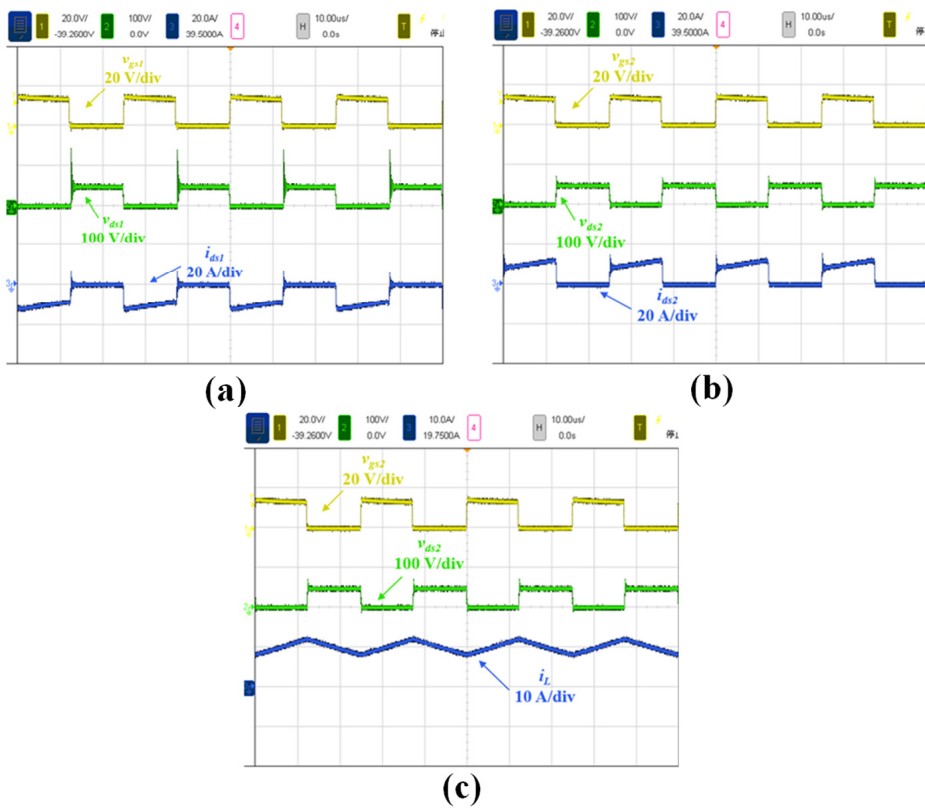

**Figure 19.** Measured waveforms of stage two: (**a**) v$_{gs1}$, v$_{ds1}$, and $i_{ds1}$; (**b**) v$_{gs2}$, v$_{ds2}$, and $i_{ds2}$; (**c**) v$_{gs2}$, v$_{ds2}$, and $i_L$.

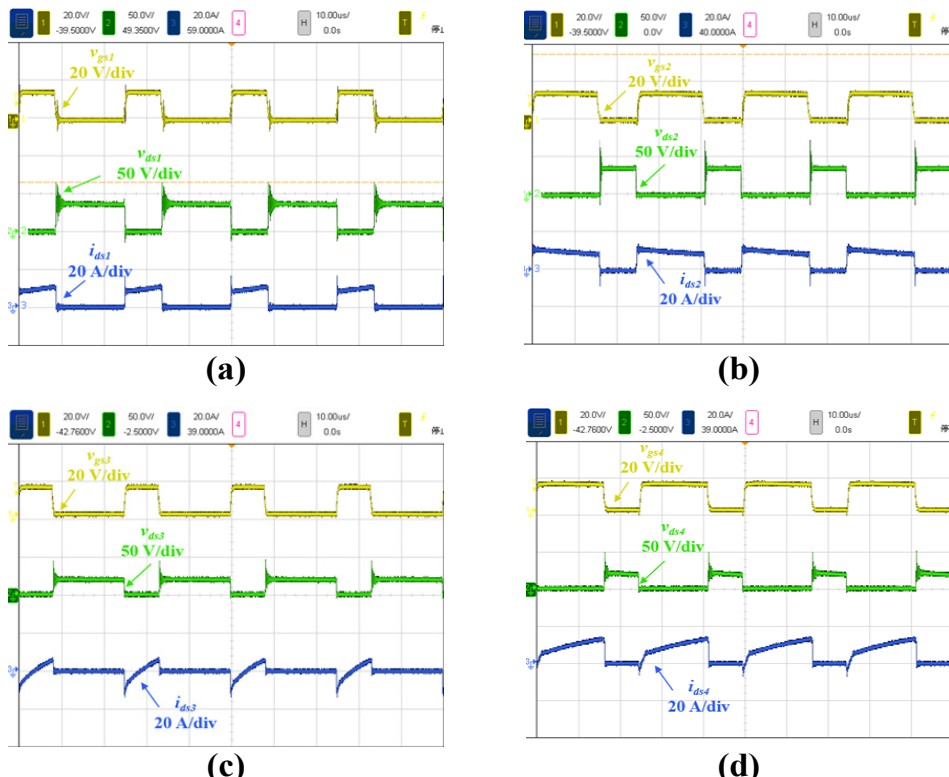

**Figure 20.** *Cont*.

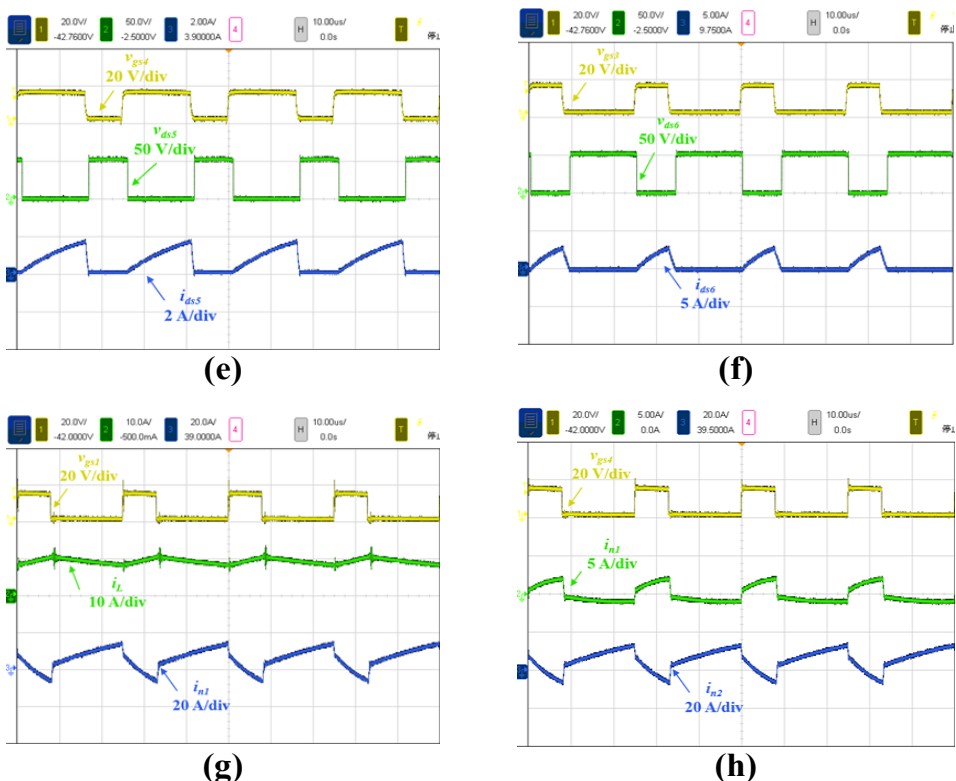

**Figure 20.** Measured waveforms of stage three: (**a**) $v_{gs1}$, $v_{ds1}$, and $i_{ds1}$; (**b**) $v_{gs2}$, $v_{ds2}$, and $i_{ds2}$; (**c**) $v_{gs3}$, $v_{ds3}$, and $i_{ds3}$; (**d**) $v_{gs4}$, $v_{ds4}$, and $i_{ds4}$; (**e**) $v_{gs4}$, $v_{ds5}$, and $i_{ds5}$; (**f**) $v_{gs3}$, $v_{ds6}$, and $i_{ds6}$; (**g**) $v_{gs1}$, $i_L$, and $i_{n1}$; (**h**) $v_{gs1}$, $i_{n1}$, and $i_{n2}$.

Figure 21a–g show the measured waveforms of stage four operating under a full load (500 W). Figure 21a shows the measured waveforms ($v_{gs1}$, $v_{ds1}$, and $i_{ds1}$) of switch *S1*. It can be seen that inductor *L* released energy through switch *S1* at this time. Figure 21b shows the measured waveforms ($v_{gs2}$, $v_{ds2}$, and $i_{ds2}$) of switch *S2*. It can be seen that inductor *L* stored energy through switch *S2*. Figure 21c shows the measured waveforms ($v_{gs3}$, $v_{ds3}$, and $i_{ds3}$) of switch *S3*. Current $i_{ds3}$ started to rise from positive to negative, and the leakage inductance current $i_{Lk}$ released and recovered energy through switch *S3*. The energy was transferred to the magnetizing inductance $L_m$. Figure 21d shows the measured waveforms ($v_{gs4}$, $v_{ds4}$, and $i_{ds4}$) of switch *S4*. Figure 21e shows the measured waveforms ($v_{gs5}$, $v_{ds5}$, and $i_{ds5}$) of switch *S5*. Figure 21f shows the measured waveforms ($v_{gs6}$, $v_{ds6}$, and $i_{ds6}$) of switch *S6*. Figure 21g shows the measured waveforms $v_{gs1}$, the transformer primary-side current $i_{n1}$, and the transformer secondary-side current $i_{n2}$.

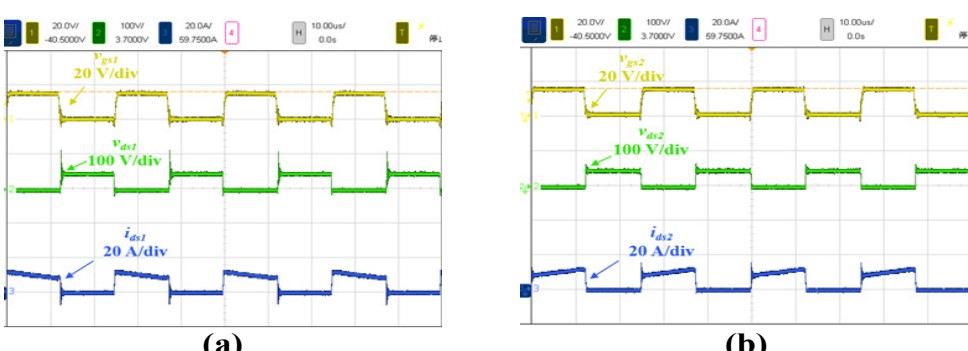

**Figure 21.** *Cont.*

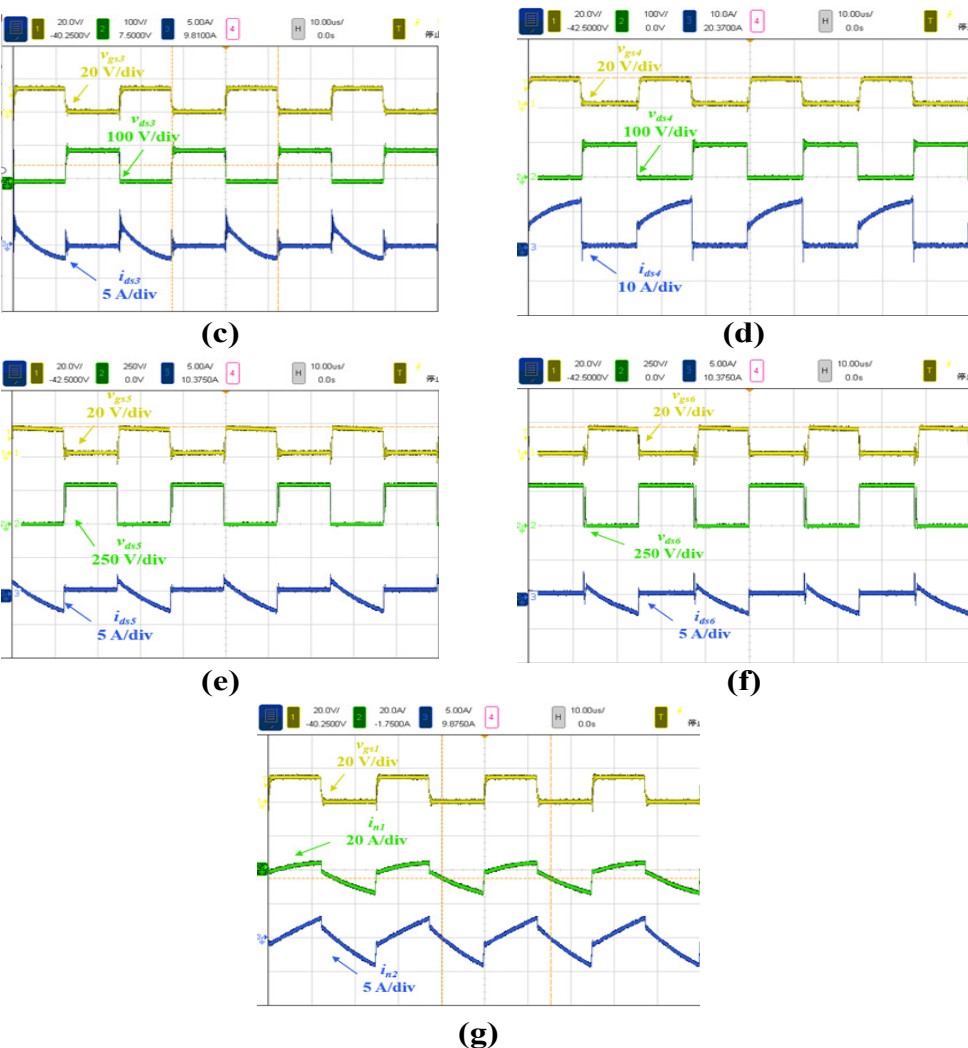

**Figure 21.** Measured waveforms of stage four: (**a**) v$_{gs1}$, v$_{ds1}$, and $i_{ds1}$; (**b**) v$_{gs2}$, v$_{ds2}$, and $i_{ds2}$; (**c**) v$_{gs3}$, v$_{ds3}$, and $i_{ds3}$; (**d**) v$_{gs4}$, v$_{ds4}$, and $i_{ds4}$; (**e**) v$_{gs5}$, v$_{ds5}$, and $i_{ds5}$; (**f**) v$_{gs6}$, v$_{ds6}$, and $i_{ds6}$; and (**g**) v$_{gs1}$, $i_{n1}$, and $i_{n2}$.

Figure 22 shows the load variation in the step-up mode. When the output load was step changed from a half load to a full load, it can be seen that the output voltage ($V_H/V_{Bat}$) was stable and not greatly affected by load changes.

Figure 23a–d show the efficiency curves of the proposed converter in this paper. Figure 23a shows the highest efficiency was 95.5% at an output power of 200 W, and the efficiency at the full load of 500 W was 92.3%. Figure 23b shows the highest efficiency was 97.8% at an output power of 150 W, and the efficiency was 95.7% at the full load of 500 W. Figure 23c shows the highest efficiency was 94.5% at an output power of 150 W, and the efficiency was 87% at the full load of 500 W. Figure 23d shows the highest efficiency was 93.4% at an output power of 150 W, and the efficiency was 88.3% at the full load of 500 W.

Figure 24 presents the power loss analysis of the proposed converter operated in step-up mode (stage three) and under full load. It can be seen that the switching loss and conduction loss of all power switches account for a large proportion of the overall power loss.

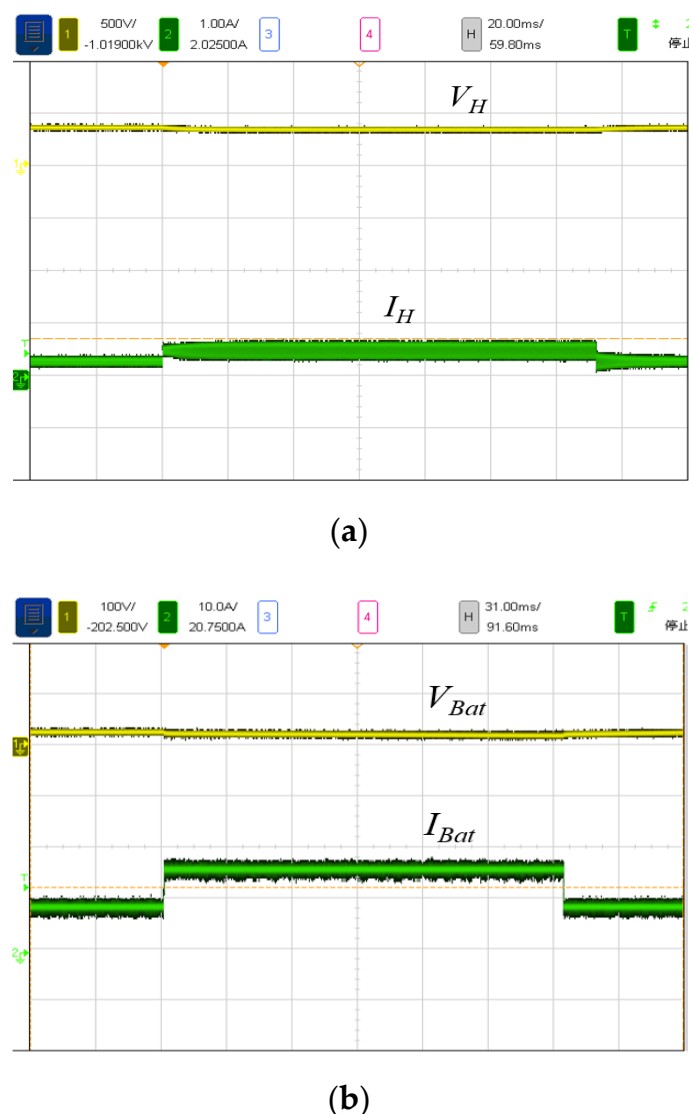

(**a**)

(**b**)

**Figure 22.** Step variation of the output load of the proposed topology: (**a**) step-up mode; (**b**) step-down mode.

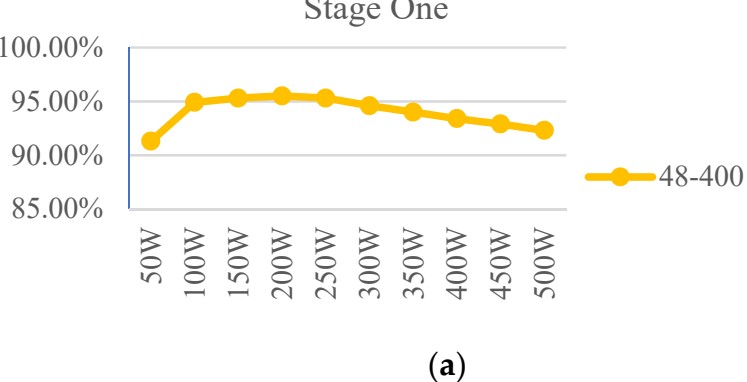

(**a**)

**Figure 23.** *Cont.*

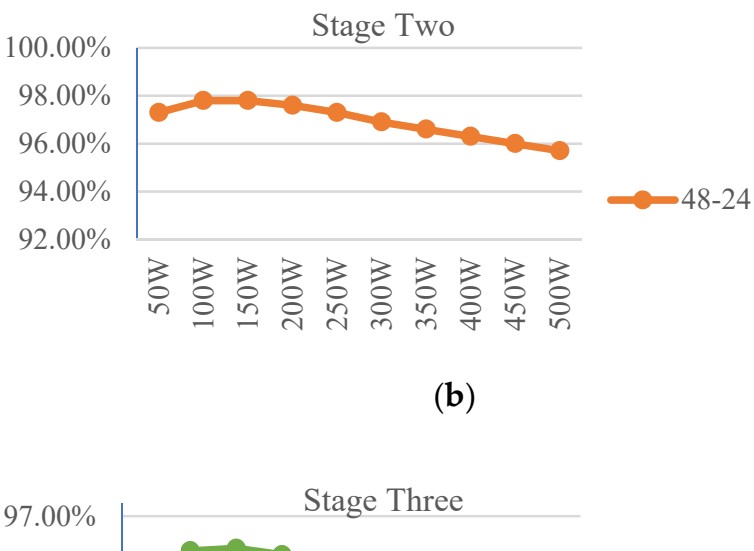

(**b**)

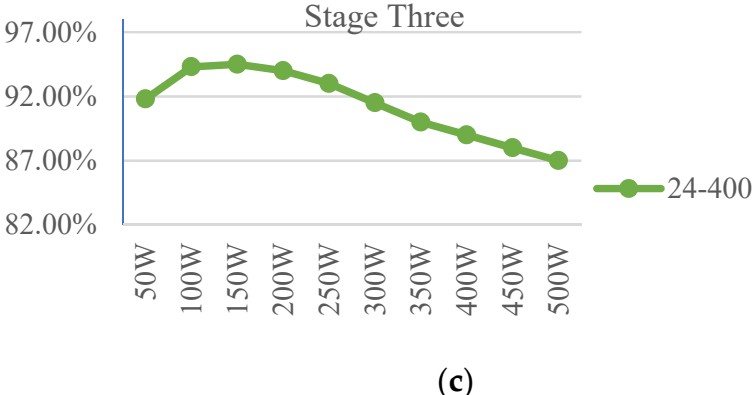

(**c**)

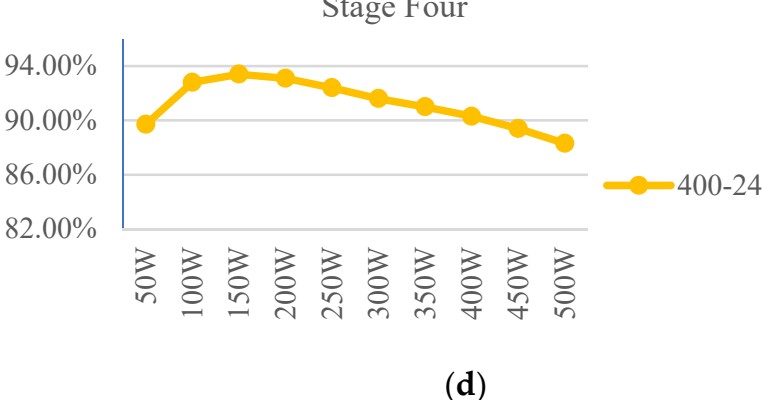

(**d**)

**Figure 23.** Efficiency curve of the proposed converter: (**a**) stage one; (**b**) stage two; (**c**) stage three; (**d**) stage four.

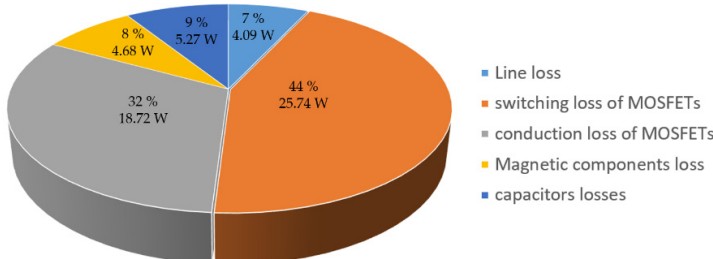

**Figure 24.** Power loss analysis of the proposed converter operated in step-up mode (stage three).

Figure 25 presents the power loss analysis of the proposed converter operated in step-down mode (stage four) and under full load. It can be also seen that the switching loss and conduction loss of all switches account for a large proportion of the overall power loss.

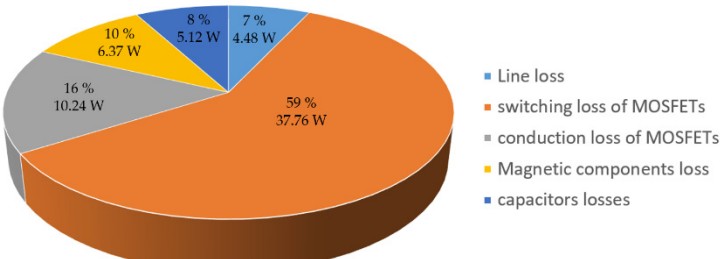

**Figure 25.** Power loss analysis of the proposed converter operated in step-down mode (stage four).

Figure 26 shows the efficiency comparison between this paper and Refs. [15–19] operating under the step-up/step-down mode. Refs. [15,16] employed more passive components, causing poor efficiency. Ref. [18] had higher conversion efficiency in the step-down mode, but its disadvantage was a low voltage gain ratio. Ref. [19] had the highest efficiency under the step-up mode, but its implemented power was only 300 W, and the total component count was too high, which limited its application. Overall, the proposed topology had a higher voltage gain ratio and efficiency compared to Refs. [15–19].

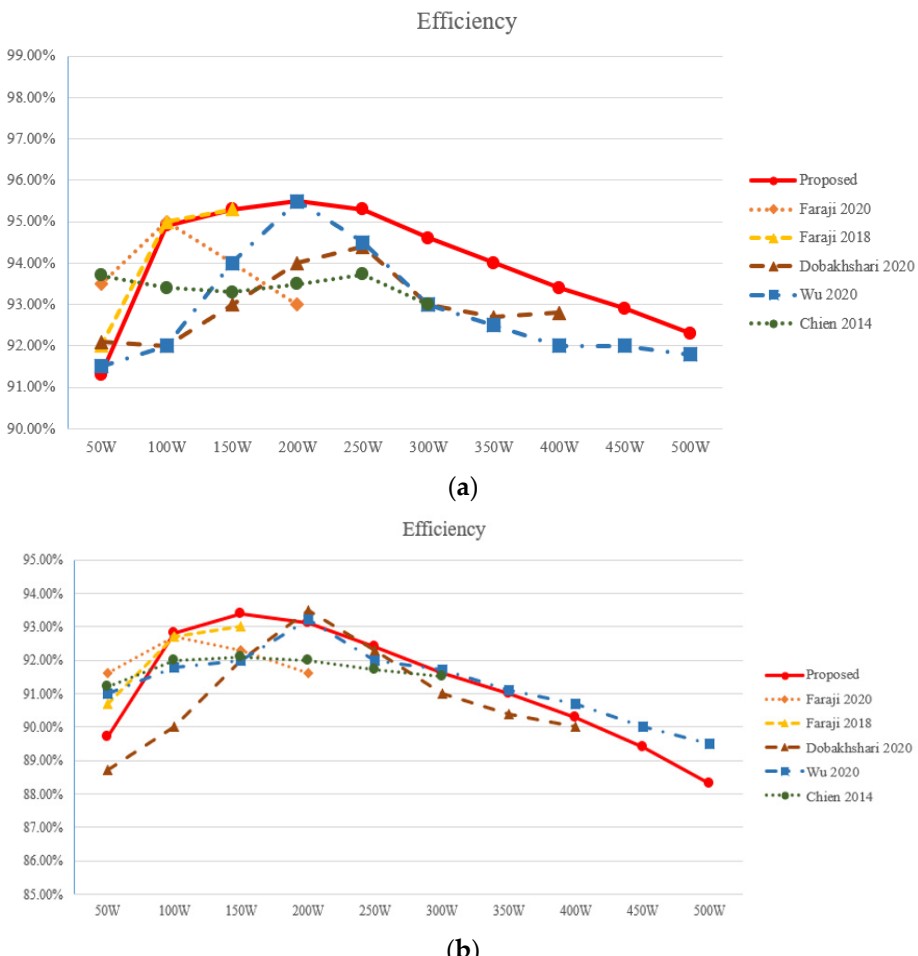

**Figure 26.** Comparison of efficiency: (**a**) step-up mode; (**b**) step-down mode [15–19].

To illustrate the feasibility and performance of the proposed converter, Table 2 shows the performance comparison between the proposed converter and Refs. [15–19]. It can be seen that the proposed topology possessed a higher voltage conversion ratio, and the implemented power was also higher than that found in other literature. Only one set of complementary PWM signals was used in each mode, ensuring that there were many stages, and the control was simple. A galvanic isolation feature was used to protect the circuit operation.

**Table 2.** Comparison of related literature of three-port bidirectional converters.

| Item/Ref. | [15] | [16] | [17] | [18] | [19] | Proposed Converter |
|---|---|---|---|---|---|---|
| Voltage of input/battery/output (V) | 48/40/400 | 30/48/400 | 30/24/400 | 24/24/200 | 24/48/400 | 48/24/400 |
| Rated power | 200 W | 150 W | 200 W | 500 W | 300W | 500 W |
| MOSFETs | 3 | 4 | 3 | 4 | 3 | 6 |
| Diodes | 7 | 6 | 3 | 3 | 5 | 0 |
| Capacitors | 4 | 5 | 8 | 4 | 6 | 5 |
| Magnetic elements | 2 | 2 | 4 | 2 | 2 | 2 |
| Efficiency of step-up mode | 92.9% | 95.3% | 95.5% | 95.3% | 94% | 94.5% |
| Voltage gain of step-up mode PV-DC bus | $\frac{1+N}{1-D}$ | $\frac{1+N}{1-D}$ | $\frac{N}{1-D}$ | $\frac{N}{1-D}$ | $\frac{1+N}{1-D}$ | $\frac{N}{(1-D)D}$ |
| Voltage gain of step-down mode DC bus-Battery | $\frac{1-D}{1+N}$ | $D$ | $\frac{1-D}{N}$ | $\frac{1-D}{N}$ | $\frac{1-D}{1+N}$ | $\frac{(1-D)D}{N}$ |
| Switching frequency | 100 kHz | 50 kHz | 100 kHz | 50 kHz | 50 kHz | 40 kHz |
| Isolated | NO | NO | Yes | Yes | NO | Yes |
| Operational mode | 3 | 4 | 3 | 3 | 3 | 4 |

## 5. Conclusions

This paper proposed a novel three-port bidirectional DC–DC converter. The steady-state analysis of the component design and the experimental results verified the feasibility and practicability of the proposed converter. The proposed converter had the following advantages: (1) a simple structure; (2) only one set of complementary signals used in each stage, providing simple control; (3) leakage inductance recovery; (4) multiple operating stages, which improved the practicability of the proposed converter; and (5) an electrical isolation feature to protect the low-voltage input side. Finally, in terms of the measured efficiency, the highest efficiency of the PV output stepped up to the high-voltage-side mode was 95.5%, and the highest efficiency of the PV terminal stepped down to the battery was 97.8%. The highest efficiency was 94.5% when the battery was stepped up to the high-voltage side, and the highest efficiency was 93.4% when the high-voltage side was stepped down to the battery.

**Author Contributions:** Conceptualization, Y.-E.W.; methodology, Y.-E.W. and R.-R.H.; formal analysis, Y.-E.W. and R.-R.H.; investigation, R.-R.H.; resources, Y.-E.W.; writing—original draft preparation, Y.-E.W.; writing—review and editing, Y.-E.W.; project administration, Y.-E.W.; funding acquisition, Y.-E.W. All authors have read and agreed to the published version of the manuscript.

**Funding:** This research received no external funding.

**Institutional Review Board Statement:** Not applicable.

**Informed Consent Statement:** Not applicable.

**Data Availability Statement:** Not applicable.

**Conflicts of Interest:** The authors declare no conflict of interest.

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
