# Peer review of "Multi-Functional Isolated Three-Port Bidirectional DC/DC Converter for Photovoltaic Systems"

_sustainability, doi:10.3390/su141811169_

Round 1

Reviewer 1 Report

1. The format of the article should be consistent, such as font, left and right alignment, etc.

2. Some figures are unclear, such as Figures 18-21.

3. Adding more relevant journal papers to the references in the past two years is recommended.

4. It can be seen from Table 2 that more power switches are used in this paper, which will increase the circuit's design cost and affect the circuit's efficiency.

Author Response

  1. The format of the article should be consistent, such as font, left and right alignment, etc.

Author response:

Thank you for reviewer’s comments, the format has been modified in the revised paper.

  1. Some figures are unclear, such as Figures 18-21.

Author response:

Thank you for reviewer’s comments. Figures 18-21 have been improved.

  1. Adding more relevant journal papers to the references in the past two years is recommended.

Author response:

Thank you for reviewer’s comments, two papers (Ref. 20-21) for three-port bidirectional DC/DC converter have been added in the revised paper.

  1. It can be seen from Table 2 that more power switches are used in this paper, which will increase the circuit's design cost and affect the circuit's efficiency.

Author response:

Thank you for reviewer’s comments, it can be seen from Table 2 that there are six power switches used in this paper, but no diodes are used. In terms of the number of semiconductor components used, the number is the smallest. Although the efficiency is not optimal, it is also moderate. Besides, there are four operation modes in this paper, functionally superior to other papers.

Response:

Thanks for reviewer's comments and guidance. The authors try to reply to the problems of reviewer, please reviewer tolerant if there are any incomplete.

Reviewer 2 Report

Excellent work.

Line number 319; the word "some" should be replace with "same"

Author Response

Line number 319; the word "some" should be replace with "same".

Authors Response

Thanks for reviewer’s comments. It has been corrected in the revised paper.

Response:

Thanks for reviewer's comments and guidance. The authors try to reply to the problems of reviewer, please reviewer tolerant if there are any incomplete.

Reviewer 3 Report

Explain the principle of converter efficiency. What is the efficiency of the converter that has been developed?

Author Response

Explain the principle of converter efficiency. What is the efficiency of the converter that has been developed?

Authors Response

Thanks for reviewer’s comments. Figures 24 and 25 present the power losses analysis of the proposed converter operated at step-up mode and step-down mode, respectively. It can be seen that the switching loss and conduction loss of all power switches account for a large proportion of the overall power loss.

Response:

Thanks for reviewer's comments and guidance. The authors try to reply to the problems of reviewer, please reviewer tolerant if there are any incomplete.

Reviewer 4 Report

1.     In the proposed circuit, the battery is kept in between about 20% to 80% SoC for reliability reasons. When the battery voltage is variable upon its SoC, it is important to see the characteristics of this circuit in case the input voltage changes.

(1) Give the allowable battery voltage range for the proposed converter in Table 1.

(2) Briefly discuss the effects of battery voltage variation on the design of magnetic component, especially for the transformer.

2.     As shown in Figure 24, the circuit efficiency measured in boost mode appears to be higher than that in buck mode totally. Comment on possible power loss differences between buck and boost modes to explain this result.

Author Response

  1. In the proposed circuit, the battery is kept in between about 20% to 80% SoC for reliability reasons. When the battery voltage is variable upon its SoC, it is important to see the characteristics of this circuit in case the input voltage changes.

(1) Give the allowable battery voltage range for the proposed converter in Table 1.

(2) Briefly discuss the effects of battery voltage variation on the design of magnetic component, especially for the transformer.

Author response:

Thank you for reviewer’s comments. When the battery voltage fluctuates, the main influence on the converter is the magnetic components and the duty cycle. If the input voltage (Vbat) is too low or too high than the design voltage, the turns ratio or inductance calculation will be insufficient, which will cause voltage drop or malfunction. If the battery voltage variation is not large, its voltage error can be corrected by adjusting the duty cycle of converter. Therefore, the magnetic components should be designed based on the average operating voltage of the battery (Vbat). The allowable battery voltage range for the proposed converter is 20V-28V.

  1. As shown in Figure 24, the circuit efficiency measured in boost mode appears to be higher than that in buck mode totally. Comment on possible power loss differences between buck and boost modes to explain this result.

Author response:

Thank you for reviewer’s comments, Figures 24 and 25 present the power losses analysis of the proposed converter operated at step-up mode and step-down mode, respectively. It can be seen that the switching loss and conduction loss of all power switches account for a large proportion of the overall power loss.

Response:

Thanks for reviewer's comments and guidance. The authors try to reply to the problems of reviewer, please reviewer tolerant if there are any incomplete.

Reviewer 5 Report

This paper proposed the multi-functional isolated three-port bi-directional Dc/Dc converter for PV systems. The measured results show the accuracy of the proposed three-port power topology. There are some requirements could be improved.

1.   The circuit diagram and text label in the paper need to be consistent, such as solar voltage VPV, battery voltage VB and high voltage output VH.

2.  The title of the submitted paper is the application of PV system. If the photovoltaic system needs to add MPPT, how to realize it?

3.   The time units for the x-axis should be displayed for the measured results.

Author Response

This paper proposed the multi-functional isolated three-port bi-directional Dc/Dc converter for PV systems. The measured results show the accuracy of the proposed three-port power topology. There are some requirements could be improved.

  1.  The circuit diagram and text label in the paper need to be consistent, such as solar voltage VPV, battery voltage VBand high voltage output VH.

Authors Response

Thanks for reviewer’s comments. It has been modified in the revised paper.

  1. The title of the submitted paper is the application of PV system. If the photovoltaic system needs to add MPPT, how to realize it?

Authors Response

Thanks for reviewer’s comments. The bidirectional converter proposed in this paper can use MCU software design to realize the function of MPPT. In order to avoid the over length of the paper, this part is not discussed in this paper.

  1. The time units for the x-axis should be displayed for the measured results.

Authors Response

Thanks for reviewer’s comments. The time units have been added in the measured figures.

Response:

Thanks for reviewer's comments and guidance. The authors try to reply to the problems of reviewer, please reviewer tolerant if there are any incomplete.